# Hydrogen Bonds: Raman Spectroscopic Study

**DOI:** 10.3390/ijms22105380

**Published:** 2021-05-20

**Authors:** Boris A. Kolesov

**Affiliations:** A.V. Nikolaev Institute of Inorganic Chemistry, Siberian Branch, Russian Academy of Sciences, 630090 Novosibirsk, Russia; kolesov@niic.nsc.ru

**Keywords:** Raman spectra, strong hydrogen bonds, tautomerism, proton tunneling, proton hopping

## Abstract

The work outlines general ideas on how the frequency and the intensity of proton vibrations of X–H···Y hydrogen bonding are formed as the bond evolves from weak to maximally strong bonding. For this purpose, the Raman spectra of different chemical compounds with moderate, strong, and extremely strong hydrogen bonds were obtained in the temperature region of 5 K–300 K. The dependence of the proton vibrational frequency is schematically presented as a function of the rigidity of O-H···O bonding. The problems of proton dynamics on tautomeric O–H···O bonds are considered. A brief description of the N–H···O and C–H···Y hydrogen bonds is given.

## 1. Introduction

An interaction between atomic and molecular systems can be divided into three main types: Coulomb, van der Waals, and chemical. Chemical interactions are usually divided into several types to make them more definite: Covalent, ionic, donor-acceptor, hydrogen, etc.; however, each of them is based on the well-known interaction of atomic or molecular orbitals. The different degree of overlap of the orbitals and the distribution of electron density on them (the occupancy of the orbitals) reveals the specified structuring of the chemical interaction.

The Coulomb and van der Waals interactions are isotropic, and the chemical one is directional. This is their main difference. The consequence of directionality is the variety of structures that arise during chemical interaction. The hydrogen bond energy is about an order of magnitude higher than the van der Waals energy, but is the weakest among all types of chemical bonds.

In recent years, general questions concerning the nature of hydrogen bonding have been much less discussed in the literature. Modern quantum-chemical programs are excellent at calculating molecules and crystal fragments, including those that involve a hydrogen bond. The result of the calculation is usually a description of the molecular orbitals and the vibrational spectrum of the compound. This is very important information. However, researchers are often interested in issues related to trends and functional dependencies, the main factors influencing certain characteristics of compounds to make possible to predict their properties from their composition and structure. This fully applies to the assessment of the characteristics of hydrogen bonds. In this brochure, an attempt is made to discuss issues related to the general principles of hydrogen bond formation. Knowledge of these principles makes our work conscious and allows for meaningful experimentation.

The hydrogen bond, being, in fact, a conventional chemical bond, is significantly inferior to the latter in strength (energy), but even more significantly surpasses it in a variety of properties. The length of a hydrogen bond of the same type can vary over a broad area, and its spectroscopic parameters in vibrational spectra vary so much that reliable assignment of the bands in the spectrum to hydrogen bond vibrations is still a difficult problem.

The importance of hydrogen bonds in chemistry and nature cannot be overestimated. Suffice it to point out that the structure of liquid water and ice, perhaps the most essential compounds for life on Earth, is determined by O–H···O hydrogen bonds. Moreover, their organization and mobility in liquid water are so complex that the structure of the latter is still a subject of discussion. All complex biological objects that make up living organisms (including, of course, people) are united by hydrogen bonds, mainly N–H···O. The hydrogen bonding is an element of the structure of most organic compounds.

Over the past 50–70 years, numerous attempts have been made to describe hydrogen bonds, mainly in terms of their structural parameters and manifestation in vibrational (IR and Raman) spectra. A large number of monographs and reviews on hydrogen bonds in various systems have appeared [1,2,3,4,5,6,7,8,9,10]. However, the authors’ desire to give an exhaustive description of the structural parameters and spectra of all compounds that form the full range of hydrogen bond properties led to excessive detail in the description of experimental data, complicating a clear presentation of the subject of research and its understanding. The complexity of the definition (or classification) of the hydrogen bond lies in the fact that it can involve various groups of atoms of the periodic table (oxygen, nitrogen, carbon, halogens, metals), and in some cases, individual molecular orbitals. Furthermore, in each case, the properties of the bond become deeply individual. In addition, the simplest (and most common) hydrogen bond consists of three atoms: Donor, acceptor, and hydrogen. Further, if the structural position of the donor and acceptor is determined with acceptable accuracy, then the position of the hydrogen atom on the bond is often not established at all but is set by the experimenter. However, it is the latter parameter that determines the spectral properties of the hydrogen bond. It is therefore clear that for many years, right up to recent years, hydrogen bonding constituted a difficult (almost incomprehensible) problem.

The purpose of this work is not a detailed description of each of the huge number of hydrogen bond manifestations, but an exposition, close to phenomenological, of the conditions and features of the occurrence of hydrogen bonds in a wide range of their energies.

It is assumed that the reader has a basic understanding of the hydrogen bond, therefore, many of its non-fundamental details are not considered here: Single- and multicenter bonds, bond angle, etc. Moreover, in order to focus the reader’s attention on the main physical and chemical aspects of the hydrogen bond, and not on all possible manifestations of it, further in the text we will mainly consider the O–H···O bond as the most widespread and studied.

## 2. Definition, Brief History, Main Stages of Development

According to the IUPAC, 2011 recommendation “The hydrogen bond is an attractive interaction between a hydrogen atom from a molecule or a molecular fragment X–H in which X is more electronegative than H, and an atom or a group of atoms in the same or a different molecule, in which there is evidence of bond formation” [11].

This strict definition is already fraught with the above-mentioned tendency to excessive detail and is perceived upon first reading with some difficulty. For this reason, readers are offered here another, possibly imperfect, but simpler and more compact definition: *The hydrogen bond is a weak chemical bond between the X–H group of one molecule and the electronegative Y atom of another* (*or the same*).

Usually, a hydrogen bond is written as X–H···Y, where the dots represent the hydrogen bond itself. In this notation, X–H is a terminal fragment of one molecule and Y is a terminal fragment of another molecule. X is a hydrogen bond donor and Y is an acceptor. The electronegativity of an element is not constant and depends on the specific chemical conditions (for more details, see next section).

The hydrogen bond was discovered at the beginning of the last century. It is not possible to establish the exact date of its discovery, since it is not clear from the publications of those years what exactly the authors had in mind when describing chemical reactions involving the polar group X–H. For the first time, the concept of hydrogen bonding appeared in the work of Huggins [12] and almost simultaneously in Latimer and Rodebusch [13]. The term “hydrogen bond” was first used by L. Pauling in 1930 [14]. However, real interest in hydrogen bonding arose only a few more years later, when Bernal and Fowler [15] suggested that a high degree of short-range order in the structure of liquid water is provided by intermolecular bonds, which were then attributed to hydrogen bonds. This work, which assumes a tetrahedral environment for each water molecule due to interactions with neighboring molecules, presently known as the “Bernal-Fowler rule”, served as a powerful impetus for the study of hydrogen bonds. However, in those years, practically the only experiment providing information on the state of the hydrogen bond in a compound was IR absorption spectroscopy. Nevertheless, in the IR (and then in Raman) spectra, the hydrogen bond is well recorded only for weak bonds and very unreliably for moderate and strong ones. For this reason, until about the 1990s of the last century, there was only a quantitative accumulation of information on the hydrogen bond, which allowed J. Pimentel and O. McKellan to declare in their famous monograph [2]: “Currently, the H-bond theory is the subject of considerable controversy, its qualitative predictive power is limited, and it can hardly make quantitative predictions. Apparently, it can be argued that the significance of the H-bond in the general theory of chemical bonding is not yet understood in all details“.

The next stage in the study of the theory of hydrogen bonding was the work of Morokuma [16], whose appearance in 1971 was due to the development of reliable computer methods for calculating molecular states. In the proposed approach, presently known as Morokuma decomposition, the chemical bond between atoms is broken down into separate components (electron repulsion, charge transfer, polarization, dispersion forces), from which the total interaction is calculated as a function of the distance between atoms. Morokuma’s work did not become the last in the development of the theory of hydrogen bonding, but dispelled (and this is its value) numerous attempts to attribute some unusual properties to the hydrogen bond. Morokuma showed that the interactions that form a hydrogen bond make it virtually indistinguishable from any other chemical bond, with the only exception that all these interactions in a hydrogen bond are relatively weak.

After the appearance of Morokuma’s work, it was long believed that hydrogen bonding occurs due to the lone pair of the acceptor, which spreads to the filled X–H orbital of the donor. However, first Benoit and Marx [17] in 2005, and then Wang et al. [18] in 2014 using a high-level quantum-chemical calculation (the work in [18] was also reinforced by the experiment data) showed that an important role in the formation of hydrogen bonds is played by the quantum uncertainty of the position of the proton. In fact, these last two works have finally established the nature of the hydrogen bond, after which various interpretations in its definition should disappear. It is the ideas [17,18] about the role of quantum uncertainty in the coordinates of atoms participating in the formation of hydrogen bonds that underlie the interpretation of its nature, adopted in this brochure.

## 3. General Description

In a conventional chemical bond, the distance between interacting atoms can vary within small limits, while the interaction potential changes monotonically and predictably. In a hydrogen bond, a change in the donor–acceptor distance *d*_X···Y_ causes a dramatic change in the potential function of the proton on the bond. This is the basis of the complex manifestation of hydrogen bonds in vibrational spectra, which for many years made it difficult to understand its nature.

Therefore, to solve the main issues related to hydrogen bonding, it is necessary to consider how the potential function of the proton on the bond, the strength of the hydrogen bond, the O–H vibration frequency, and other characteristics change depending on *d*_O···O_.

A huge amount of experimental material has been accumulated in the literature characterizing weak and moderate hydrogen bonds. There are much fewer works dealing with strong and extremely strong bonds, and the Raman spectral studies of compounds with such bonds presented in this work were performed for the first time. This is due to the fact that, as a rule, fixing of strong hydrogen bonds in vibrational spectra requires measurements at low temperatures. Such measurements, while not presenting fundamental difficulties, do require, however, expensive equipment and time-consuming, and therefore are not widespread. However, first, let us briefly consider the general properties of hydrogen bonds.

### 3.1. Potential Energy of a Proton on a Hydrogen Bond

In a weak O–H···O hydrogen bond, the proton practically does not interact with the oxygen-acceptor, but is fully localized at the oxygen-donor, forming a strong ordinary covalent bond with the latter.

Therefore, the potential function of the proton along the hydrogen bond at large values of *d*_O···O_ > 2.7 Å can be represented as a curve with two minima, one deep near the donor, and the other shallow near the acceptor (Figure 1). Both vibrational states, zeroth and the first excited (denoted in Figure 1 as “0” and “1”), are located in a deep minimum.

A decrease in the *d*_O···O_ distance enforces the interaction of a proton with an acceptor. In this case, the energy minimum for a proton near the acceptor deepens, and for the intermediate H-bond (*d*_O···O_ ~ 2.6 Å) the potential curve for a proton looks like in Figure 2.

The main difference between the moderate hydrogen bond and the weak (and, as we will see below, from the strong and extremely strong) is that now the proton potential is not harmonic, and the known solutions of the vibrational problem for the harmonic potential cannot be attributed to the case. In the potential of the moderate hydrogen bond, the excited vibrational state can be located above the barrier separating the minima near the donor and acceptor, i.e., in a wide potential well formed by both minima, and the zeroth vibrational state is in a narrow minimum near the donor atom. In this case, the ratio of the energy of the stationary states, zeroth and the first excited, can differ appreciably from that of traditional solution for a harmonic oscillator.

With a further decrease in *d*_O···O_ and the transformation of the H-bond from moderate to strong, the interaction of the proton with the acceptor becomes so strong that it is compared with its interaction with the donor, while the energy minima for the proton near the donor and acceptor become identical, leaving a barrier with a height of *U*_0_ between them (“deep tunneling regime”, Figure 3). This situation formally means that the electronegativity of the donor and the acceptor is equal, but in reality, most often, we are talking about the complete identity of the donor and acceptor molecules. The condition of identity is necessary for the formation of a strong hydrogen bond, but not sufficient. The second condition is the high electronegativity of the donor and acceptor.

If *U*_0_ turns out to be less than the energy of zero-point vibrations of the proton in the bond (more on this below), then the zero vibrational state, as well as the excited state, gets a wide minimum, and double-well potential of the proton becomes single-well.

Finally, as the donor and acceptor approach further, an extremely strong hydrogen bond is formed (“ultrashort, centered HB”, Figure 4), in which there is practically no barrier between the minima, and the shape of the potential curve of the proton becomes close to harmonic. The latter case is a linear symmetric hydrogen bond O···H···O. However, the preparation of compounds with linear symmetric bond is not an easy task from the point of view of chemistry.

### 3.2. The Bond Energy as Function of its Length. Uncertainty of the Proton Coordinates, Uncertainty of the O···O Distance

Benoit and Marx [17] and then Wang et al. [18], using the procedure for calculating of the Feynman path integrals for the O–H···O system, determined the positions of the proton on the H-bond for the case of weak, moderate, strong, and extremely strong hydrogen bonds. The main result of these works is that the bonded proton is considered not as a mathematical point, but as the proton density distribution, which is the result of the quantum uncertainty of the proton coordinates—an indisputable fact, but not explicitly taken into account in earlier theoretical works (Figure 5).

Due to quantum uncertainty, the proton density distribution function, having a finite width comparable to the potential space for a proton at a bond, begins to spread with decreasing *d*_O···O_ from a minimum at a donor atom to a minimum at an acceptor atom. In other words, part of the proton density turns out to be near the acceptor oxygen and interacts with it according to the same scheme as with the donor oxygen (Figure 6).

It is this effect that causes, on the one hand, the strengthening of hydrogen bonds, and on the other, a decrease in the frequency of the ν_O-H_ vibration (for more details on the mechanism of formation of the proton vibration frequency, see the next paragraph 3.3). The closer the donor and acceptor are to each other, the greater part of the proton density is in the potential well of the acceptor, the stronger the interaction of the proton with the acceptor, and the tougher the hydrogen bond. At large values of *d*_O···O_, the proton density distribution function does not reach the adjacent minimum (case (a) in Figure 5). However, a weak hydrogen bond occurs in this case as well. What is the matter here?

Of course, the primary reason for this is the deformation of the potential curve of the proton in the presence of an acceptor atom and, as a consequence, a decrease in the energy of stationary states of the oscillator (for more details, see the next section). However, our understanding of position uncertainty also turns out to be useful in this case. Indeed, it can be shown that the value of the standard deviation of the coordinate can be written as
(1)$$\delta q\left(0\right)=\sqrt{\frac{\hbar }{2m\omega }}=\sqrt{\frac{\hbar }{2m\sqrt{\frac{k}{m}}}}\;∼m^{-1/4}.$$

An oxygen atom is 16 times heavier than a hydrogen atom, hence the standard deviation δ*q*(0) and the quantum uncertainty of its coordinate Δ*x* are two times less than that of a proton. In other words, the uncertainty of the position of the proton on the hydrogen bond doubles, as it were, due to the uncertainty of the position of oxygen atoms.

The particle momentum and the quantum uncertainty of the particle momentum in the oscillator are proportional to its energy $p=\sqrt{2mE}.$ Hence, the uncertainty of the proton’s coordinate is inversely proportional to its binding energy (see uncertainty relation) and thus depends on the rigidity of the hydrogen bond. The energies of the vibrational states of the proton, zero and first excited, are maximal for the weak and extremely strong bonds (Figure 1 and Figure 4, respectively), and minimal for the moderate and strong bonds corresponding to the initial phase of the transition from the double-well potential to the single-well potential (Figure 2 and Figure 3). From the uncertainty relation, it follows that the maximum uncertainty of the proton coordinate should fall on the case presented in Figure 2 and Figure 3, and the minimum—in Figure 1 and Figure 4. It is the pattern that manifests itself in the calculations of Benoit and Marx [17] (Figure 5).

Thus, the energy of the hydrogen bond depends on the degree of propagation of the proton density to the neighboring (acceptor) minimum and is determined by the donor–acceptor distance. However, neither the depth of the minimum at the acceptor, nor *d*_O···O_ are independent parameters, but are derived from another quantity—the electronegativity of the acceptor, i.e., the ability of the acceptor to take on the electron density and, thus, interact with the hydrogen atom. The latter value is determined by the degree to which the atomic orbitals of the oxygen acceptor are filled when molecular orbitals are formed in the “acceptor” molecule. If the oxygen–acceptor orbitals are completely saturated (theoretically), then the oxygen loses the ability to interact with the hydrogen atom of another molecule and the hydrogen bond is not formed. Furthermore, the strongest O–H···O hydrogen bond occurs when the oxygen-acceptor bond with “its” molecule is close to an ordinary bond, and the oxygen needs an additional electron density to saturate its orbitals. In this case, the saturation of the oxygen-donor and oxygen-acceptor are the same and the resulting hydrogen bond becomes symmetrical. Thus, the entire spectrum of the strength (energy) of the O-H···O hydrogen bond is determined by the electronegativity of the oxygen acceptor, which determines the corresponding length of the *d*_O···O_ bond.

### 3.3. The Proton Vibrational Frequency as Function of a Length of the Hydrogen Bond

With the formation of a hydrogen bond, even a weak one, the resulting additional minimum near the acceptor deforms the potential curve for the proton near the donor so that the minimum becomes asymmetric and wider. A formal consequence of the broadening of the potential well is a decrease in the frequency of the proton vibration, since the latter value is proportional to the force constant *k*, determined by the slope of the potential well: *k* ~ *d*^2^*U*/*dx*^2^. However, the dependence of the proton vibrational frequency on the distance *d*_O···O_ is complex, and it is necessary to consider it in more detail.

From the text of the previous paragraph, it follows that the penetration of the proton density into the neighboring potential well near the acceptor enforces the hydrogen bond H···O and weakens the O–H bond, since both are carried out by the valence electron of the same hydrogen atom. For a long time, determining the dependence of the frequency ν_O–H_ on the distance *d*_O···O_ was a priority goal in the study of a hydrogen bonding. There is no need for it now, since, as already mentioned above, the vibrational frequency ν_O–H_ of any hydrogen bond can be calculated with a high accuracy. As general remarks on the dependence of ν_O–H_ on *d*_O···O_ the following can be given. The change in ν_O-H_ with the strengthening of the hydrogen bond can be monotonic only as long as both vibrational states, the zeroth and the first excited, are placed in the same potential minimum near the donor, which is valid in the range of bond strength from weak to moderate. However, at *d*_O···O_ ~ 2.6 Å, the moment comes when the excited state is “pushed out” from a narrow minimum into the region of a wide potential well formed by the combination of two potential minima (Figure 2). At this point, the energy of the excited state drops sharply due to an increase in the size of the oscillator and the corresponding lengthening of the de Broglie wavelength of the proton.

In fact, the wave functions describing the vibrational states of the harmonic oscillator are found in the solution of the Schrodinger equation and for the zeroth and first excited states are written as
(2)$$\psi_{0}\left(q\right)=\frac{1}{N_{n}}e^{-\frac{\beta^{2}q^{2}}{2}},$$
(3)$$\psi_{1}\left(q\right)=\frac{2\beta q}{N_{n}}e^{-\frac{\beta^{2}q^{2}}{2}}$$
where *N_n_* is the normalizing factor, *q* is the coordinate, and $\beta =\frac{1}{q_{max}\left(0\right)}$.

According to de Broglie, a quantum particle is also a wave, the length of which in the case of rectilinear movement of the particle is determined by its momentum *p*:(4)$$\lambda =\frac{h}{p}$$

The wave functions shown in Figure 7 are de Broglie waves. Zeroth state is represented by a half-wave, and the first excited is represented by one full period of the wavelength of λ determined by the size of the oscillator (the wave must “fit” in the oscillator, and that is the substance of the boundary conditions). From here,
(5)$$p=\frac{h}{\lambda }=\sqrt{2mE},$$
(6)$$E=\frac{h^{2}}{2m\lambda^{2}}∼\frac{1}{\lambda^{2}}.$$
where *E* is the energy of the vibrational state. It should be noted that a comparative estimation of the energy of the vibrational state by the de Broglie wavelength is possible only within alone stationary state, for example, only for the zeroth state, or only for the first excited state. In addition, as already noted above, the shape of the potential curve of the oscillator, shown in Figure 2, is far from harmonic, and this circumstance will radically change the solution both for the energy of the stationary states and for the ratio between the energies of the zeroth and first excited states.

When the excited state is pushed out from a narrow minimum to a broad one (Figure 2), the de Broglie wavelength of the excited proton increases by 1.5–2 times, and the energy of the excited state decreases by 2–4 times compared to what it would be in the absence of a wavelength jump (Figure 8). At the same time, the zeroth state remains in a narrow minimum and the size of its potential space and energy remain virtually unchanged. An indirect confirmation of this assumption can be found in the calculations of Wang et. al. [18].

In other words, of the two states, *E*(0) and *E*(1), only the second one is greatly reduced due to the unusual shape of the potential curve. As a result, the vibrational frequency ν_O–H_ = *E*(1) − *E*(0) also strongly decreases, and ν_O–H_ as a function of *d*_O···O_ should fall sharply in the region of *d*_O···O_ ~ 2.6 Å. (Figure 9). This is the most dramatic moment in the evolution of hydrogen bonding.

The sharp decreasing of the ν_O–H_ is possible only by the fact that the potential shown in Figure 2 is not harmonic, it does not include the known solutions for the eigenvalues of energy in a quantum harmonic oscillator, and the relationship between the energy of stationary states, according to which the energy of the zeroth state in a harmonic oscillator is *ħ*ω/2, and the energy of the first excited state is 3*ħ*ω/2, is violated. However, the idea of the wave function in the form of a de Broglie wave “inscribed” in the size of the oscillator remains in this case as well. 

The next stage on the dependence of the vibrational frequency OH occurs at a distance of *d*_O···O_ ~ 2.5 Å (Figure 3). At this distance, the barrier between the potential minima becomes less than the energy of zero-point motions, and now the zeroth state is pushed out into a wide minimum of united potential. The potential for a proton takes the traditional single-well form, the ratio of the energies of the zeroth and excited states also becomes close to the usual one, and the difference between them is set to *ħ*ω. Starting from this point, the width of the potential well narrows as the *d*_O···O_ decreases, and the ν_O–H_ dependence should again display a monotonous change, but this time in the direction of higher frequencies (Figure 9).

It is possible to estimate what the distance *d*_O···O_ should be, so that the wavenumber of a proton vibrations in a single-well potential again takes the value about 3600 cm^−1^, as in the case of a very weak hydrogen bond. The vibrational frequency of a proton of a weak O–H···O bond is determined by the O-H force constant, while the vibration of a strong symmetric O···H···O is determined by two identical O···H force constants. Therefore, in order for ν_O–H_ in both types of potential to be equal, each of the force constants O···H of the symmetric bond must be twice as small as one force constant O–H of the weak bond. However, if the force constant of O–H turns out to be half, then the frequency will decrease to ~ 2500 cm^−1^. The last value corresponds to the length of the O–H ~ 1.1 Å. Hence, the proton vibration in the single-well potential of the extremely strong O···H···O bond with a frequency of ~3600 cm^-1^ will take place at *d*_O···O_ ~ 2.2 Å. 

To experimentally confirm the dependence of the proton vibration frequency on the *d*_O···O_ distance, presumably determined by the curve in Figure 9, it is necessary to gradually reduce the *d*_O···O_ distance in the same compound. It is possible with the application of external pressure. Such an experiment was carried out when measuring the vibrational spectrum of ice under outer pressure [19,20]. It was shown that at a pressure of about 60 Gpa, the spectrum of ice changes radically: The mode of stretching vibrations of H_2_O at ~1500 cm^−1^ (according to [19]) or ~2700 cm^−1^ (according to [20]) disappears, but instead a new mode about 800 cm^−1^ appears. The latter rises to ~ 1500 cm^−1^ with a further increase in pressure. In addition, the Raman intensity of the O–H vibrations drops to zero, while the IR intensity, on the contrary, increases with a very strong broadening of the absorption band. It can be seen that the changes in the Raman spectra observed in [19,20] at outer pressure are quite consistent with the scenario described above. It should be noted, however, that the authors themselves interpreted this as a phase transition ice VII—ice X. Therefore, in view of the results [19,20], the wavenumber ~800 cm^-1^ should be taken as the lower limit of the proton vibration frequency on the O–H···O bond, and this minimum value should be attributed to a moderate hydrogen bond, and not to a strong or extremely strong one, as assumed earlier [21]. Thus, the arrangement of the vibrational states of a proton in the potential of a moderate hydrogen bond, when the zeroth state remains in a narrow minimum at donor oxygen, and the first excited state falls into a wide minimum (Figure 2), causes the fundamental change in the Raman spectra: A sharp and significant decrease in the frequency of O–H and a weakening in the scattering intensity to zero.

The experiment with ice [19,20] perfectly illustrates the behavior of the O–H···O hydrogen bond as a function of the *d*_O···O_ length, but the O–H···O bond itself in liquid and crystalline water is not a conventional O–H···O bond. The fact is that the oxygen atom of the same H_2_O molecule is both a donor and an acceptor of the hydrogen bond simultaneously, whereas in all other compounds, the oxygen-donor and oxygen-acceptor are physically different atoms. What does this change in the state of the O–H···O bond? First of all, the rate at which the bond strength changes with a change in the *d*_O···O_ distance. In an isolated H_2_O molecule, the electron shell of the oxygen atom is saturated due to two ordinary O–H covalent bonds and its electronegativity is close to zero. For this reason, the O–H···O hydrogen bond formed in liquid and crystalline water is very weak with a proton vibrational frequency around of 3200–3400 cm^−1^. Due to the weak hydrogen bond, the melting point of ice is very low and we have life on Earth as it is. When the *d*_O···O_ length is shortened, for example, under external pressure, and the hydrogen bond is strengthened, the hydrogen atoms begin to share its electron density with neighboring oxygen atoms, as a result of which their bond with “their” oxygen weakens, and the saturation of the electron shell of this oxygen from “their” protons decreases, provoking an increase in its electronegativity. Hence, the electronegativity of the oxygen atom is variable and increases with decreasing *d*_O···O_. In other words, the donor and acceptor properties of the H_2_O molecule in liquid water or in a crystal increase strictly at the same rate with a decrease in *d*_O···O_. This trend is not limited by anything and, it would seem, by increasing the external pressure, one can achieve an extremely strong hydrogen bond in crystalline H_2_O with a proton frequency of 3000–3600 cm^−1^. However, this is not the case. The H_2_O molecule in both the liquid and the crystal is surrounded by four other molecules and forms two donor and two acceptor bonds. However, four hydrogen bonds can only be equivalent to two ordinary covalent bonds of an oxygen atom, one ordinary bond of oxygen as a donor, and one bond of the same oxygen atom as an acceptor. In other words, the energy of an ordinary covalent bond is divided in liquid or crystalline water between two protons, limiting the maximum possible frequency of their vibrations at low *d*_O···O_ by value of 2000–2500 cm^−1^.

To estimate the height of the barrier *U*_0_, for which a transition from moderate to strong bonds can occur, it is necessary to establish an energy criterion for the transition. Comparison of *U*_0_ with the value of thermal energy *kT*, which is usually used when comparing energy states, is not applicable in this case, since, in the case of vibrations, it is valid only for temperatures at which vibrational states with a large quantum number *n* are excited (otherwise, “ultraviolet catastrophe”). The frequency of the proton vibrations on the hydrogen bond is high, so the excitation of vibrations with a large quantum number is possible only at high temperatures and is not realized in compounds with hydrogen bonds. Therefore, the criterion for the transition from a double-well potential to a single-well potential is the ratio of the barrier height *U*_0_ and the energy of zero-point vibrations of the proton. The latter value at low temperatures turns out to be much higher than the energy *kT*. Figure 3 shows a situation in which the energy of zero-point vibrations begins to exceed the value of the barrier *U*_0_. It is this case that signifies the initial phase of the transition of the double-well potential of the proton to the single-well potential, or, in other words, the transition from a moderate bond to a strong one. The use of *kT* as a transition criterion can yield a significant underestimation of the distance *d*_O···O_ at which this transition occurs.

It should also be noted that the relationship between the strength of the hydrogen bond, the width of the potential minimum of the proton, the de Broglie wavelength, and the energy of the vibrational states can be used to estimate the vibrational frequencies of a proton on a hydrogen bond of any rigidity. In other words, the main effect of hydrogen bonding, i.e., the dependence of the proton vibrational frequency on the *d*_O···O_ distance, is a consequence of the change in the width of the potential minimum of the proton and the corresponding change in the de Broglie wavelength.

### 3.4. Half-Width of the O–H···O Vibrational Bands

When the O–H···O hydrogen bond changes from a weak to moderate one, the bandwidth of the O–H vibration increases very much, and its peak intensity often weakens so that the band becomes barely noticeable in the spectrum. Among several reasons for the band broadening considered in the literature, there is one that is reliably confirmed experimentally.

The bandwidth of the vibrational band is inversely proportional to the lifetime of the excited vibrational state and the hydrogen O–H···O bond is determined by the interaction of the high-frequency O–H vibration with other modes, including low-frequency crystalline modes. Crystal vibrations yielding a modulation of the O···O distance are translational optical vibrations of hydrogen-bonded molecules relative to each other. The latter are the vibrations of the hydrogen bond itself. Their frequency for medium-sized molecules (for example, organic molecules with a mass of 50–100 at. units) lies in the region of ~100 cm^−1^, i.e., well below the ν_O–H_ frequency. This means that the frequency of the ν_O–H_ stretching vibration can follow the change in the *d*_O···O_ distance, which results from the excitation of intermolecular vibrations. In other words, two modes, the high-frequency valence ν_O–H_ and the low-frequency phonon mode of lattice vibrations, interact with each other, and this interaction is proportional to the interaction parameter χ = Δν/Δ*d*, i.e., the slope of the ν_O-H_ dependence on the *d*_O···O_ distance.

As an example, Figure 10 shows the diagram of the hydrogen bonds of the H_2_O molecule and the OH-groups in the cavity of the hemimorphite mineral, Zn_4_Si_2_O_7_(OH)_2_·H_2_O, and Figure 11a shows the spectra of stretching vibrations of H_2_O molecules and OH-group in the cavities of hemimorphite at 4 K and 300 K, demonstrating the change in the width of the vibrational modes with temperature. In this case, the temperature controls the population of low-frequency crystalline modes. In fact, the frequencies of crystal phonons are usually in the range of 50–300 cm^−1^. The Boltzmann population of such phonons varies from 0 to 1 in the temperature range 4–300 K.

The higher the population of phonon states, the greater the probability of interaction of high-frequency O–H vibrations with crystal modes, the shorter the lifetime of the excited state, and the larger the mode bandwidth. This explains both the strong increase in the bandwidth with temperature and this bandwidth itself. The fact that the active interaction of the O–H stretching vibration with the lattice phonons actually takes place can be seen from Figure 11b, where the region of the combined modes in the Raman spectrum of hemimorphite is shown in close-up, i.e., scattering by combinations of ν_O–H_ with lattice phonons (in this case, with translational vibrations of the H_2_O molecules themselves and the lattice of the host crystal).

### 3.5. Intensity of O–H···O Vibrational Bands

It is well known that when the wavenumber of ν_O–H_ of the O–H···O hydrogen bond becomes less than 2700 cm^−1^, the intensity of the band in the Raman spectra decreases to almost zero, and the band itself becomes extremely inexpressive, consisting of several maxima with intensity at the spectrum noise level (see, for example, [23]). Finally, in many compounds with the moderate bond, the O–H band is not observed at all.

We have already discussed above that for the moderate hydrogen bond, one can arise the situation, shown in Figure 2, in which the excited vibrational state is pushed out into the wide united minimum and the zeroth state remains in the narrow minimum. In this case, the proton appears in different positions in the ground state and in the excited state: The unexcited proton is located in a narrow minimum next to the one of the oxygen atoms while the excited proton is located in the center of the bond. Therefore, the excitation of vibrations requires the proton to be displaced along the bond. However, in the Raman spectroscopy, the incident electromagnetic radiation interacts with the electron shell of the atoms rather than with their nuclei, since the frequency of the incident radiation (~2 × 10^4^ cm^−1^) is an order of magnitude higher than the frequency of nuclear displacement (~2 × 10^3^ cm^−1^). In IR absorption spectroscopy, the frequency of the incident electromagnetic radiation must be equal to the frequency of the fundamental mechanical vibrations of the molecule to excite the vibrational quantum. In Raman spectroscopy, the frequency of incident radiation is high, much higher than the frequencies of mechanical vibrations of atoms; therefore, the field of an incident electromagnetic wave causes a displacement not of atoms, but only of their electron shells. This results in the appearance of Hertzian dipoles and Rayleigh, i.e., frequency upshifted, scattering. However, the reaction of the electron density to an external field depends on the polarizability of chemical bonds, which is modulated by mechanical vibrations. The latter are slow compared to the frequency of the incident radiation. It is the modulation of the polarizability of the system during the vibrations that causes the appearance of Raman scattering at the vibrational frequency of a molecule or crystal. Thus, the difference between IR absorption and Raman scattering is that the incident radiation interacts with vibrating atoms in the first case and with their electronic subsystem in the second. As a result, the processes accompanied by atomic displacements are not revealed in the Raman spectra but can be active in IR absorption spectra. In other words, the damping of the scattering intensity on O–H vibrations for the moderate bonds is a result of the shape of the potential curve (Figure 2), at which the proton, passing from the zeroth vibrational state to the first excited state, is forced to change its position on the bond. This circumstance greatly complicates the study of the properties of the moderate hydrogen bond and, possibly, served as a source of errors in the past, when some bands in Raman spectrum were assigned to O–H vibrations of the moderate hydrogen bond without sufficient experimental justification. As we will see below (see Section 4 and Section 5), the intensity of the scattering is restored as soon as the energy of both states, zeroth and the first excited, becomes higher the barrier between potential wells and the coordinate of the proton ceases to depend on its vibrational state again.

### 3.6. How Does Single-Well Potential Occur?

The reasons and conditions for the formation of a single-well potential and the characteristics of the latter are quite simple, but we will briefly discuss them, since in Chapter III, the width of the potential well and the height of the barrier will become the most needed parameters when interpreting experimental results.

There are three factors that determine the *d*_O···O_ distance, the barrier height *U*_0_, and, as a consequence, the hydrogen bond strength.

Identity of a donor and an acceptor, which in practice simply means the formation of a hydrogen bond by two identical molecules. If the molecules are different, then the interaction of the proton with the donor will always be stronger than with the acceptor, and the O–H length is shorter than H···O. It prevents the formation of the minimum *d*_O···O_ and strong hydrogen bond. When the molecules are completely identical, the concepts “donor” and “acceptor” lose their meaning.Electronegativity of oxygen-donor and oxygen-acceptor. It was already mentioned above that electronegativity determines how strongly an oxygen atom interacts with a proton, i.e., how close a proton can come to an oxygen atom, both a donor and an acceptor.Thermal vibrations. Thermal vibrations increase the distance between atoms due to anharmonic processes. In a molecular crystal, the vibrational spectrum starts from 20–40 cm^−1^. Consequently, the excitation of equilibrium thermal vibrations starts from 40–60 K. In other words, the minimum possible hydrogen bond length can be established only at T ≤ 40–60 K.

In the harmonic approximation, the particle energy is proportional to the square of the deviation of the particle coordinate from the equilibrium position, *U*(*x*) = *kx*^2^, where *k* is the force constant. If the minimums in the double-well potential are deep enough, then each of them can be considered as harmonic. Then, for a double-well symmetric potential, in which the middle of the length between the donor and the acceptor is chosen as the origin, and the distance from the origin to each of the minima is denoted as ±Δ, the potential function of the proton is written as follows
*U*(*x*) = {[*k*(*x* + Δ)^2^]·[*k*(*x* − Δ)^2^]}^1/2^ = ±*k*(*x* + Δ)(*x* − Δ)(7)

Hence, at *x* = 0, the height of the barrier *U*_0_ (choosing a positive value)
*U*_0_ = *k*Δ^2^,(8)

The force constant depends on the electronegativity of the oxygen atoms and does not change for a given compound (the exception is the H_2_O molecule, see above). Distance Δ depends on the external pressure and the temperature of the crystal, which determines the population of thermal vibrations. In other words, with an increase in external pressure or a decrease in temperature, the barrier height *U*_0_ decreases; therefore, it is easy to obtain the situation shown in Figure 3, at which both vibrational states are within the wide united minimum and the scattering becomes active.

## 4. Experimental Study of Strong Hydrogen Bonds

Chemical compounds with a strong and extremely strong O–H···O hydrogen bond, in the spectra of which the transition to a single-well potential is confidently detected, are quite rare. In addition, the transition itself, as a rule, is stretched in temperature, and begins at T < 150 K (the reasons for this will become clear from the subsequent text). In other words, a detection of the transition to a single-well potential requires the Raman measurements in the temperature range 5–300 K, which is not a widespread practice. In this work, Raman spectra of benzoic acid crystals, where the early stage of the transition to a single-well potential is observed, glycine phosphate crystals with the stable state of the single-well potential (Figure 3), and dimethylphormamid [(DMF)_2_H]_2_ with the extremely strong hydrogen bond (Figure 4) were obtained. Consider the spectra of these compounds in the sequence presented.

### 4.1. The Features of the Vibrational Spectrum of Benzoic Acid

Despite the fact that benzoic acid relates to tautomeric compounds (more about them in Section 5), at low temperatures the benzoic acid dimers in the crystal are distorted due to crystalline effects (one of the four oxygen atoms of the τ-cycle has an additional short contact with the environment in the crystal lattice) so that one O–H···O bond becomes shorter than the other (Figure 12), and it is these short bonds that show in the spectra the signs of the transition to a single-well potential.

Figure 13 shows the Raman spectra of crystals of benzoic acid in the high-frequency region of O–H and C–H stretching vibrations. At low temperatures (T ≤ 60 K), a series of narrow and weak in intensity lines appears in the spectra—a phenomenon that is not observed in other H-bonded compounds. In addition, also at temperatures below 60 K, a narrow line appears in the spectrum at 2910 cm^−1^, the intensity of which becomes very high (higher than the intensity of the C–H stretchings) at a temperature of 10–15 K. With a further decrease in temperature from 10 to 5 K, the intensity of both the series of weak lines and the 2910 cm^−1^ line begins to decay.

The distance in wavenumbers between weak narrow lines is not constant and does not form any regular series; therefore, the entire series cannot be interpreted as resonant repetitions of a combination of O-H vibrations and any mode (or group of modes). However, upon closer examination, it can be found that the lines in the emerging low-temperature series are located at approximately the same intervals as the modes of crystalline and intramolecular vibrations in the range of 50–1700 cm^−1^.

Figure 14 shows the spectra of intramolecular (bottom) and high-frequency (top) vibrations at 10 K. One can see that weak narrow lines in the high-frequency spectrum (2400–4000 cm^−1^) are located and separated from each other in the same order as the modes of the intramolecular vibrations (50–1700 cm^−1^). In other words, the spectrum of weak modes in the high-frequency region is indeed a second-order spectrum, consisting of the combined tones of the hydrogen-bonded stretching vibration O–H and intramolecular vibrations. The O–H stretching itself is not recorded, but its frequency should fall within the range of 2600–2650 cm^−1^.

Due to the distortion of the dimer, the lengths of two O–H···O hydrogen bonds become different when the crystal temperature decreases. In this case, the predicted O–H stretching mode at ~2600 cm^−1^ is evidently related to the shortest O-H···O hydrogen bond, and 2910 cm^−1^–to the longer one. The emergence of the second-order spectrum means that those intramolecular vibrations that modulate *d*_O···O_ distance become capable of a very strong interaction with the О–Н valence mode of the short hydrogen bond, much stronger than, for example, the well-known interaction of lattice phonons modulating the O···O distance in a crystal with common (weak and moderate) O–H···O hydrogen bonds. Earlier (see paragraph 3 and Figure 9), it was noted that at the moment when the excited vibrational state is pushed out to the wide minimum of the unified potential, the wavenumber of the proton vibration should sharply decrease due to an increase in the size of the oscillator. At this, even a negligible modulation of the O···O distance can cause a significant change in the ν_O–H_ frequency. It is the phenomenon that causes the very strong interaction of the corresponding intramolecular modes with the O–H stretching vibration, the interaction that provokes the emergence of a second-order spectrum. In benzoic acid crystals at low temperatures, the length of the short O–H···O bond does not yet reach critical values and the excited vibrational state is not yet pushed into the wide minimum, but those intramolecular vibrations, which modulate the length of the short O–H···O bond, produce the conditions for this length to fall into the zone of a rapid drop in the proton vibration frequency and, as a consequence, a strong interaction between intramolecular and O–H vibrations of the short hydrogen bond. This phenomenon is able to produce soliton states in crystals (Davydov’s solitons). No changing in the peak position of the second-order bands and the mode at 2910 cm^-1^ was observed at different temperature. This probably means that the arising new state of the short hydrogen bond is not stable but exists only as a result of the vibrational interaction.

Thus, the transition of the potential from double-well to single-well in benzoic acid crystals is not complete and is characterized by its initial stage, which arises only when the intramolecular modes modulate the bond length. This is quite understandable, taking into account that the *d*_O···O_ distance in benzoic acid dimers is rather long, ~ 2.6 Å, and for a complete transition to the single-well potential, a bond length less than 2.5 Å is required.

As will be shown in paragraph 4.2 of this chapter, the vibrational frequency of a proton in the steady-state single-well potential of a strong hydrogen bond is very low, ~ 930 cm^−1^, while the ν_O–H_ frequency in benzoic acid immediately before the transition to the single-well potential is much higher—around 2600 cm^-1^. This means that a discontinuity in the dependence of ν_O–H_ on the *d*_O···O_ distance should be indeed observed, as it was suggested above (see Figure 9).

### 4.2. Strong Hydrogen Bonds. Glycine Phosphate

Glycinium phosphite crystal, C_2_H_8_NO_5_P, contains in its structure two short O-H···O hydrogen bonds with a length of 2.49 Å and 2.53 Å [25,26,27,28]. The oxygen atoms in each of these bonds are identical and the proton potential is described at room temperature as a symmetrical double -well potential. Figure 15 shows the fragment of the crystal structure.

IR spectrum of glycinium phosphite at ambient temperature is shown in Figure 16. Figure 17a,b show the Raman spectra of glycinium phosphite in the range 900–1000 cm^−1^ at various temperatures (a) and at different polarization of incident and scattered light relative to the crystallographic axes.

A strong hydrogen bonding in glycinium phosphite arises between P–O bonds of the PO_3_H_2_ anions (Figure 15). These bonds have different lengths, 1.513 and 1.529 Å at room temperature, and their vibrations appear in the Raman spectrum (Figure 17a,b) as two lines, one of which, the low-frequency mode at 964 cm^−1^, is assigned to stretching vibration of O–P–O fragment with the maximum contribution from the long P–O bond, and the other, high-frequency mode at 971 cm^−1^, is assigned to the O–P–O vibration with the predominant contribution from the short P–O bond. (Here and below, the vibrational wavenumbers refer to a temperature of 5 K). An assignment of the Raman bands related to the stretching vibrations of the main chemical bonds in the crystal was made on the basis of a quantum chemical calculation presented in [29]).

With temperature decreasing, the relative integral intensity of the 964 and 971 cm^−1^ modes decrease and two other modes at lower frequency, 939 and 943 cm^−1^, arise (Figure 17a). In addition, a broad structureless band in the region of 930–980 cm^−1^, shown by the dashed line, also appears in the spectrum. The shape of the latter is completely unusual: The maximum intensity of the band falls on its low-frequency edge, i.e., at ~ 930 cm^−1^, and the minimum is at high-frequency (i.e., approximately at 980 cm^−1^).

Figure 17b shows the polarized spectra of the crystal at 5 K, obtained for different crystallographic directions. It can be seen that a broad band appears only in the *ab*-plane of the crystal, in which both O-H···O hydrogen bonds lie. The relative integral intensity of the broad band as a function of temperature is shown in Figure 18.

Short O-H···O bonds in the crystal form infinite chains along the *a* axis; however, the bonds themselves in the chains alternate in length and direction (Figure 15). Since the oxygen atoms in each bond are pairwise completely identical to each other, the protons in each pair must be able to occupy positions with equal probability both at one oxygen atom and another. The identity of the oxygen atoms creates chemical conditions for the formation of a strong hydrogen bond, while the potential function of the protons at room temperature is symmetric double-well. At the same time, both oxygen atoms belonging to the same PO_3_H_2_ group cannot simultaneously act as acceptors, since this creates a deficiency in the valence of phosphorus atoms in this group. Thus, at a low temperature, a conflict situation should arise, in which the protons, on the one hand, should be ordered along the entire length of the O–H···O bond chain (i.e., over the entire crystal), and, on the other hand, they should be distributed equiprobably near both oxygen atoms in each pair. This contradiction disappears when the protons on each of the O-H···O bonds occupy the position in the midpoint of the bond. The last condition means the formation of a single-well potential.

The theoretical calculation fulfilled in [17] and the consideration presented in the previous sections of this work show that the *d*_O···O_ distance around of 2.5 Å or less provides the formation of a single-well potential. The *d*_O···O_ distance of 2.43 Å was called in [17] as threshold for the transformation of the pronon potential to a single-well and the thermal energy *kT* at low temperatures served as the criterion. However, as already mentioned, the only correct criterion is the energy of zero-point motions *ħ*ω/2, which, for reasonable value of ω, is much higher than *kT*. Consequently, the threshold value of the *d*_O···O_ distance is also higher than that assumed in [17], and the *d*_O···O_ in glycinium phosphite fully ensures the formation of a single-well potential.

The interpretation of the above experimental results is as follows. At room temperature, both hydrogen bonds are symmetric double-well due to the high value of the barrier height *U*_0_. However, due to phonon-assisted jumping, the protons on the hydrogen bond are statistically distributed between both positions on the bond. The conflict between the statistical distribution of the protons and the valence ability of phosphorus atoms reveals at T > 50 K a sharp increase in the half-widths of most bands in the spectrum, and their peak positions show significant dispersion.

With temperature decreasing and freezing of the vibrations, the *d*_O__···O_ distance and the height of the barrier decrease. The appearance of two additional (to the bands at 964 and 971 cm^−1^) O–P–O stretching modes at 939 and 943 cm^−1^ (Figure 17a,b) at low temperature occurs due to the strengthening of the hydrogen bonds, i.e., to the displacement of the protons towards the center of the bond. Simultaneously with the appearance of two last modes, a broad band at 930–980 cm^−1^ arises in the spectrum (dashed curve in Figure 17a).

The band is recorded only in the *ab*-plane of the crystal and in addition demonstrates the unusual contour, which strongly differs from the Lorentzian or Gaussian contours (Figure 17a,b). Both of these arguments allow us to attribute the band to the vibrations of protons of the strong hydrogen bond.

Figure 19 shows a qualitative interpretation of an unusual contour of the proton band.

It has already been mentioned above that *d*_O__···O_ distance is not fixed and is determined by the Gaussian distribution due to the quantum uncertainty of the oxygen atom coordinates (see Section 2).

One can assume that the most probable *d*_O__···O_ distance in glycinium phosphite corresponds at low temperatures to the potential curve where the ground vibrational state of the proton coincides with the top of the barrier (Figure 19b, green curve). Then the ground state for larger *d*_O__···O_ appears lower than the potential barrier (Figure 19b, red curve) and falls within the region where the scattering on proton vibrations is forbidden. For shorter *d*_O__···O_ (Figure 19b, blue curve), both the ground and the excited states appear within the same wide potential minimum, and the scattering becomes Raman-active again. This particular circumstance is responsible for the unusual shape of the scattering band in glycinium phosphite (Figure 17a,b). The low-wavenumber edge of the broad band, where the Raman intensity is maximal, should be assigned to the proton vibrations on the O-H···O hydrogen bond with *d*_O__···O_, at which the zero-point energy achieves the top of the barrier and both the ground and the first excited states get the wide minimum. Thus, the glycinium phosphite crystal is a perfect example of that the evolution of the hydrogen bonding transfers to a new stage at which both the ground and the first excited vibrational states appear within the same wide minimum of the single-well potential.

The proton vibrations in the single-well potential of a strong O-H···O hydrogen bond was observed for the first time in Raman spectra (as far as we know). The structural parameters of the bonds [25,26,27,28] required for the transition from a symmetric double-well to a single-well potential are close to those predicted theoretically [17]. The energy of zero-point motions *ħ*ω/2 of a proton, which is estimated as ~1000 cm^−1^ from a vibration frequency of ~2000 cm^−1^ for moderate hydrogen bonds, significantly exceeds the theoretical value of the barrier between wells at short hydrogen bonds [17], which allows us to talk about the proton vibrations in a single-well potential. The unusual contour of the proton vibrational band can further serve as the most important criterion for the experimental confirmation of the potential conversion to a form in which the barrier between adjacent minima is preserved, but the energy of zero point motions of the proton exceeds its height. With a further decrease in the O···O distance of the hydrogen bond, the potential barrier decreases, the bond becomes extremely strong, and the potential well gets the traditional form of a harmonic potential (Figure 4), in which the contour of the vibrational band also turns out to be an ordinary Gaussian.

### 4.3. Extremely Strong Hydrogen Bond in [(DMF)_2_H]_2_

The [(DMF)_2_H]_2_[W_6_Cl_14_] is one of the rare compounds in chemistry, in which a very short O···H···O hydrogen bond is realized. Figure 20 shows a fragment of the structure in which oxygen atoms of DMF (N,N-Dimethylformamide) molecules can occupy two different positions with the creation of the O···H···O hydrogen bonds, the lengths of which are 2.36 Å (0.7 O1/O2 occupancy) and 2.46 Å (0.3 O1/O2 occupancy) [30] (numerical values are given for a temperature of 140 K).

The C–O bond lengths are 1.272 Å (at O···O = 2.37 Å) and 1.266 Å (at O···O = 2.474 Å). Both of the latter quantities indicate that the C–O bonds are close to ordinary bonds and therefore the shell of oxygen atoms is not saturated, and their electronegativity is high. These data, together with the complete identity of oxygen atoms, create favorable conditions for a generation of an extremely strong symmetric O···H···O hydrogen bond.

Since the *d*_O__···O_ distances in (DMF)_2_H are significantly shorter than that in glycine phosphate, the proton potential in (DMF)_2_H should be the same as shown in Figure 4, and the expected frequency of the proton vibrations is much higher as compared with glycine phosphate.

Figure 21a shows the Raman spectra the compound in the region of proton vibrations on hydrogen bonds in the temperature range 5–300 K. As the temperature decreases, two broad bands, 1370 and 1450 cm^−1^, rise in the spectra. At this, the low-frequency band appears at T ≤ 110 K, and the high-frequency band—at T ≤ 60 K. In addition, the intensity of the 1422 cm^−1^ narrow band, which is weak at room temperature, begins to increase together with the intensity of both broad bands. The wavenumber of the 1422 cm^−1^ mode is close to the vibrational frequency of the ordinary C–O bond (~1300 cm^−1^) and much lower than the wavenumber of the double C = O bond (~1800 cm^−1^). This is quite consistent with the structural data and with the assumption that the oxygen atoms involved in the emergence of the hydrogen bonds are electronegative. The intensity of the C–O vibrational mode is the result of a resonant response to the appearance of the hydrogen bonds in the Raman spectrum.

The broad bands at 1370 and 1450 cm^−1^ correspond to proton vibrations on the strong hydrogen bond. Even though this bond is characterized by a double-well potential, the barrier *U*_0_ between the minima of the latter is low due to short *d*_O···O_ distances (Figure 3). The proton vibrational frequency for such potential is directly proportional to the rigidity of the hydrogen bonding. Therefore, the modes at 1370 cm^−1^ and at 1450 cm^−1^ should be assigned to the vibrations of the longer and the shorter of these two hydrogen bonds, respectively. The temperature dependence of the intensity of these two modes is the most interesting manifestation of their features and requires a detailed discussion.

First of all, both bands appear at low temperatures. Formally, this behavior is no different from that observed in the glycine phosphate crystal. However, as we saw above, the barrier *U*_0_ between the minima is proportional to the force constant of the hydrogen bond (the slope of the potential curve), and the square of half the distance between the minima (see expression (8)). In the (DMF)_2_H dimer, the steepness of the minima is so great that even with a small lengthening of the O···O distance due to thermal vibrations of the crystal, the barrier height *U*_0_ changes sufficiently strongly to become higher than the energy of zero point motions of the proton at room temperature, i.e., in a position in which Raman scattering by proton vibrations is forbidden. With a decrease in temperature and freezing of crystalline vibrations, the hydrogen bond length becomes shorter, and the barrier becomes lower, and both vibrational states turn out to be in a wide minimum, becoming Raman-active. In other words, strong hydrogen bonding remains strong at higher temperatures, but the steepness of the potential curve and thermal vibrations make the barrier height *U*_0_ too high at room temperature in relation to the energy of zero-point motions.

For the same reason, the band at 1450 cm^−1^ is revealed in the spectrum at lower temperatures than the band at 1370 cm^−1^. This mode corresponds to the vibrations of the strongest of the two hydrogen bonds; therefore, it appears in the Raman spectrum only when all crystal vibrations, including most low-frequency modes (≤40 cm^−1^), are frozen so that the barrier is diminished until an acceptable value.

The proposed interpretation is well illustrated both by the sequence in which the bands of proton vibrations on the bonds of different rigidities appear with decreasing temperature (Figure 21a) and agrees with the assumption that the appearance or the disappearance of proton vibrational bands in the Raman spectra of moderate and strong bonds is controlled by the positions of the energies of the ground and the first excited vibrational states relative to the barrier between the minima.

The transition of the ground proton state at a low temperature from the narrow minimum next to the donor to the wide minimum of the unified potential (as it takes place in glycinium phosphite and in (DMF)_2_H) has another unexpected effect. It is well known that measuring the O···O distance by two different methods of X-ray diffraction and neutron diffraction reveals somewhat different values. This is caused by the fact that X-rays are scattered on electronic shells of the atoms while neutrons are scattered on atomic nuclei.

In the case of hydrogen bonding, the electron shell of the donor oxygen is partially shifted towards the proton due to chemical interaction with the latter. For this reason, the position of the donor atom measured in X-ray diffraction is also shifted towards the proton and to the oxygen acceptor, while the obtained *d*_O__···O_ distance is slightly underestimated, which plays no significant role in most cases.

However, as mentioned above, in the case of strong and maximally strong hydrogen bonds, the proton is localized next to the donor at high temperatures and between the donor and the acceptor at low temperatures. Therefore, the position of the oxygen donor determined by X-ray diffraction will be shifted from its true state at high temperatures and will not be shifted at low temperatures (Figure 22). At the same time, the *d*_O__···O_ value seemingly increases as the temperature decreases, while its real value can only decrease, and it was particularly observed in glycinium phosphite and in (DMF)_2_H at low temperature. This effect can be the reason for misinterpretation of data obtained from structural studies using X-ray diffraction.

## 5. Tautomeric Hydrogen Bonds

### 5.1. What Is the Proton Tautomerism?

Proton tautomerism occurs in the systems with intermolecular hydrogen bonding X–H···Y where the donor and the acceptor are indistinguishable, and the proton can jump between them. The simplest compound containing a tautomeric bond (referred to hereafter as the τ-bond) is the formic acid dimer (Figure 23). Left (*L*) and right (*R*) tautomers of this dimer are identical and constitutes a six-membered τ-ring formed by two O-H···O hydrogen bonds and two carboxyl groups.

When plotted along the direction of hydrogen τ-bonds, the proton potential function is a curve with two identical wells (Figure 24a). In the case of a symmetric double-well potential, the protons must have the opportunity of a coordinated transition between the wells, which occurs as a result of protons hopping over the potential barrier or tunneling through the barrier.

However, the symmetric double-well potential is realized only for isolated molecular formations in the gas phase. In solids, the coordinated motion of charge-carrying protons to the neighboring potential affects the interaction between a given τ-ring and the τ-rings of the neighboring molecules where this transition is not yet completed. In other words, the energies of *L*- and *R*-tautomers in the crystal lattice differ by the value *A* (Figure 24b). Therefore, the tunneling must be accompanied by the absorption of the phonon ω*_A_* (*ħ*ω*_A_* = *A*), and the temperature dependence of the tunneling rate is determined by the average value of the vibrational quantum number of phonons ω*_A_* and is anti-Stokes in nature.

The structures of compounds allowing proton tautomerism are diverse and include crystals with infinite chains, dimers, trimers, and tetramers [31]. Moreover, both the same type atoms (O-H···O, N-H···N) and different type atoms (O-H···N, N-H···O) can be donors and acceptors in tautomeric hydrogen bonds (Figure 25). Oxygen atoms in carboxylic acids, the τ-bonds of which will be considered here, are not highly electronegative. For this reason, the corresponding hydrogen O-H···O bonds are borderline between moderate and strong. Further, the nature of the symmetric double-well potential in τ-bonds differs from that considered above for the usual (not tautomeric!) O-H···O hydrogen bond.

If a τ-bond is realized by atoms of different types, the potential curve of the proton on each bond is asymmetric, as in Figure 24b, but the total energy of both tautomers is described by a symmetric curve, as in Figure 24a.

Proton transfer along τ-bonds has been broadly studied by the techniques of pulsed NMR spectroscopy and inelastic neutron scattering INS [32,33,34,35,36,37,38,39,40,41,42]. A theory of coordinated motion of hydrogen atoms along the hydrogen bonds of carboxylic acid dimers was developed by Skinner and Trommsdorff [36].

The present work is an attempt to obtain the data on the behavior of protons on the O-H···O τ-bond using simple and easily available Raman spectroscopy. For this purpose, Raman spectra of compounds with symmetric (the chains of terephthalic acid, C_8_H_6_O_4_, TPA, Figure 26a), quasi-symmetric (benzoic acid dimers, C_6_H_5_COOH, BZ, Figure 26b), and asymmetric (ibuprofen dimers, C_13_H_18_O_2_, IB, Figure 26c) τ-bonds were considered in a temperature range of 5–300 K.

The behavior of protons on the τ-bond is quite complicated and is characterized by several independent processes.

At large distances *d*_O__···O_ (>2.6 Å), a coordinated proton transfer along the τ-bond is mainly due to proton hopping over the potential barrier *U*_0_ with the participation of phonons *ħ*ω*_U_* ~ *U*_0_ (phonon-assisted hopping). The barrier height *U*_0_ in various compounds with O-H···O τ-bonds is as high as 500–1000 cm^−1^ [37]; therefore, this process is more effective at high temperatures. Back and forth transitions between *L*- and *R*-tautomers as a result of hopping do not change the length and the strength of hydrogen bonding, but they switch the positions of C–O and C = O bonds in the carboxyl groups. Therefore, the main effect of proton hopping is manifested exactly in the behavior of vibration frequencies of these two bonds.

At low temperatures, owing to the freezing of vibrations, the *d*_O__···O_ distance becomes shortest, and the barrier height becomes smallest for a given compound. In compounds with a strong hydrogen bond (*d*_O__···O_ < 2.6 Å), this makes the coordinated transfer of both protons along the τ-bond probable as a result of tunneling through the barrier. The tunneling, as well as proton hopping, does not change a force constant of τ-bond but switches the C–O and C = O bonds between themselves. The latter effect is accompanied by the changes in the lengths of C–O and C = O bonds with the tunneling rate, which is usually ~10^9^ s^−1^ [37], and the amplitude approximately an order of magnitude higher than that of equilibrium (thermal) vibrations of these bonds. Therefore, proton tunneling and proton hopping are associated with forced vibrations of C–O and C = O bonds of the carboxyl group with the result that the frequencies of their normal vibrations are subject to the anharmonic shift [43]. Thus, the temperature dependence of the C–O or C = O stretching vibrations is an indicator of both proton tunneling through the barrier and proton hopping over the barrier *U*_0_. The tunneling is significantly slowed down as a result of deuteration, since the probability of tunneling is proportional to exp (–B√*m*), where *m* is the mass of the tunneling particle and *B*—constant.

The proton distribution function in potential well is delocalized because of quantum uncertainty principle and spreads partly to neighbor (empty) potential well (Figure 27). This process has been already discussed in detail above. The smaller the *d*_O···O_ distance and the crystal temperature, the more efficient the spread of proton coordinate. When the hydrogen is replaced by deuterium, the quantum uncertainty of the deuteron on the bond decreases insignificantly, by a factor of 2^1/4^ = 1.19 (see Section 3), and it was confirmed by the calculations of Wang et al. [18].

It should be noted that, in contrast to hopping and tunneling, where a coordinated transition of protons to a neighboring well occurs in a random τ-cycle of the crystal lattice and regardless of the environment, the spread of the proton density to a neighboring potential minimum as a result of the quantum uncertainty of the proton coordinate occurs in all τ-cycles of the lattice simultaneously and to the same extent and therefore does not reveal to a change in the energy state of protons due to their interaction with neighboring cycles. In other words, the scheme shown in Figure 24b is not valid in the case of proton density delocalization.

Thus, when analyzing the experimental data, we will proceed from the following assumptions.

Coordinated proton tunneling on the τ-bond occurs mainly at low temperatures, and proton hopping occurs mainly at high temperatures;Tunneling changes the energy of their interaction with the environment and requires the participation of phonons ω*_A_* (Figure 24b);Spread of proton (deuteron) distribution function to the neighboring well increases the hydrogen τ-bond (Figure 24a);Proton tunneling and proton hopping do not change the force constant of the τ-bond but modulate the length of C–O and C = O bonds of the τ-ring;Deuteration of the τ-bond virtually does not affect the degree of proton sharing and significantly slows down the tunneling.

From this, the modes of translational vibrations of the τ-bond, ω_τ_, and the vibration of the C–O and C = O bonds of the τ-ring are most important and informative for the analysis of hopping and tunneling. The experimental assignment of spectral bands to the translational vibrations was done for all compounds using temperature dependence of corresponding frequencies and quantum-chemical calculations. In the case of terephthalic acid and ibuprofen, the assignment was additionally confirmed by the deuteration on the τ-bond [43,44]. The structural parameters of O-H···O τ-bonds and experimental and calculated frequencies ω_O···O_ are listed in Table 1.

### 5.2. Quantum Delocalization of Protons

Figure 28 shows the temperature dependences of the peak position of the translational modes for three different compounds.

The temperature dependence ω_τ_(*T*) of the symmetric τ-bond in TPA (Figure 28а) has a characteristic breakpoint whose position on the temperature scale correlates with the beginning of the freezing of translational vibrations, i.e., the temperature where the average value of the quantum number *n* of the mode at 106 cm^−1^ becomes smaller than unity. The breakpoint in the temperature dependence ω_τ_(*T*) is also observed in the crystals with a symmetric N-H···N τ-bond [22]. In the quasisymmetric τ-bond in BZ (Figure 28b), there is also a breakpoint in the ω_τ_(*T*) dependence, but it corresponds to a lower temperature than the beginning of phonon freezing at 87 cm^−1^. Finally, a breakpoint in the ω_τ_(*T*) dependence in the asymmetric case of IB (Figure 28c) is negligible and is observed at T < 50 K.

Based on ω_τ_(*T*) dependence of the symmetric τ-bond in TPA (Figure 28а), one can propose that the transition of the vibrational state of the translational mode from the excited state (*n* = 1) to the zeroth state (*n* = 0) diminishes the *x*_1_–*x*_2_ distance between potential energy minima of τ-bonds until the proton wave function starts spread to the neighboring well to increase the hydrogen bonding. As can be seen in Figure 28, the slope changing of the ω_τ_(*T*) in TPA(D) is registered at virtually the same temperature as in TPA(H). This experiment convincingly confirms the above assumption that the low-temperature change in the frequency of the translational vibration is due to the delocalization of the proton (deuteron) wave function into the neighboring well rather than its physical transfer, since in the latter case, the probability of tunneling decreases significantly with increasing mass of the tunneling particle (for more details in [49]).

Figure 28 shows that the higher symmetry of τ-bond the higher temperature of the breakpoint. It means that the degree of proton sharing is maximal for the equal proton energy states in the neighbor potential wells, that is in the case presented in Figure 24a. In principle, the breakpoint of ω_τ_(*T*) should be observed in any X-H···Y hydrogen bonds with sufficiently small X···Y distance; however, this distance itself also depends on symmetry of the given bond, i.e., in what extent the donor and acceptor are similar each other by their electronegativity.

### 5.3. Proton Hopping in Ibuprofen

IB (Figure 26c) is an example of the asymmetric τ-bond. In this case, left- and right-tautomers are initially unequal in terms of energy states and are characterized by the energy difference Δ*E*. For this reason, the ω_τ_(*T*) dependence shows very small the proton sharing (Figure 28c) and the proton tunneling should be negligible. Since the proton tunneling and the proton hopping equally affects the structure and energy of compounds, the registration of hopping in the IB spectra is greatly facilitated in the absence of tunneling.

At low temperatures, the translational vibration of the IB τ-bond is represented by a single mode at ~103 cm^−1^ corresponding to the *L*-tautomer; an additional band appears above 150 K at 93 cm^−1^ (Figure 29, shown by an arrow), which corresponds to translational vibrations of the *R*-tautomer.

Figure 30a,b shows temperature dependence of the integral intensity of these two translational modes in the original and in deuterated ibuprofen. The temperature dependences of these two intensities (the modes at ~100 cm^−1^ and at ~90 cm^−1^) show a pronounced activation character, i.e., demonstrate two τ-bond states separated by the energy Δ*E*. Solid lines in Figure 30 are plotted according to the expressions, which take into account the population of states for Δ*E* values equal to 80 and 70 MeV in original and deuterated crystals, respectively [43].

However, the true value of this experiment lies in the fact that it makes possible to reliably establish the effect of the proton transfer on the spectral characteristics of the crystal.

Figure 31b shows the peak position of the stretching C=O vibration of the carboxyl group in the ibuprofen dimer as function of temperature. When the temperature rises from 5 K, the frequency of the C = O stretchings of the *L*-tautomer (at low temperatures, only the *L*-tautomer exists) has to, on the one hand, decrease as a result of ordinary anharmonic processes, and on the other, to increase, since, due to the same anharmonic processes, the involvement of the terminal C = O group in the hydrogen OH···O = C bond weakens, which is always accompanied by an increase in frequency.

Figure 31b shows that the second process acts somewhat more efficiently and ω_C = O_ slightly increases in the temperature range 5 K–150 K. However, at T ≥ 150 K, the frequency ω_C = O_ suddenly begins to decrease exponentially, and the exponent of this process and the activation process of the *R*-tautomer is the same, i.e., ~80 MeV (Figure 30,b). In other words, the change in ω_C = O_ is strictly correlated with the appearance of the *R*-tautomer, both in terms of temperature and rate of change.

The reason for the unusual behavior of the ω_C = O_ mode is as follows. The transition from *L*- to *R*-tautomer and vice versa causes the C–O and C = O bonds of the carboxyl group to change places. In this case, an ordinary C–O bond, becoming a double, is significantly (by about 0.1 Å) shortened, and a double C = O bond is lengthened by the same amount. This process occurs with the frequency of proton hopping on the τ-bond, which in benzoic acid crystals is in the range 10^8^–10^11^ s^−1^ in the temperature range 5 K–300 K (see, for example, [39,40,41,42]). Thus, the bond lengths of the carboxyl group experience forced vibrations with frequency of hopping and very large amplitude, several times higher than the amplitude of normal vibrations. Strictly speaking, it is not vibrations, but changes of length with a great frequency. Forced vibrations of the carboxyl group give rise to the same anharmonic phenomena as normal vibrations, as a result of which the C = O bond lengthens, and the frequency of normal vibrations begins to decrease with the appearance of proton jumping, which is observed in Figure 31b. Thus, the dependence ω_C = O_ (*T*) can be used as a very sensitive tool for characterizing of proton hopping at high temperatures and proton tunneling at low temperatures in systems with τ-bond. To our knowledge, no forced vibrations of chemical bonds with any frequencies have been observed before. Consequently, this phenomenon can be attributed to a new type of forced vibrations.

The dependence shown in Figure 31b also confirms the earlier conclusion that there is no tunneling at low temperatures (i.e., in the range 5 K–150 K) in IB.

### 5.4. Proton Tunneling

The described experiment with the frequency of C=O vibrations has no particular importance for ibuprofen, since the process of proton hopping is clearly demonstrated by the dependence of the intensity of ω_τ_ translational modes in the crystal (Figure 30a,b). However, it is a very sensitive tool to characterize both proton hopping and proton tunneling to the neighboring well in other compounds such as TPA chains with symmetric τ-bonds.

As it was mentioned above, the coordinated transfer of the protons to the neighboring potential minimum should be accompanied by the absorption of the phonon ω*_A_* (*ħ*ω*_A_* = *A*, Figure 24b), and the function of the tunneling rate vs. temperature dependence is determined by the average value of the vibrational quantum number of phonons ω*_A_* and has an anti-Stokes character.

Figure 32 shows the ω_C = O_(*T*) dependences in TPA (H) and TPA (D). In this case, the anharmonic shift of the frequencies of C = O vibrations is proportional to the tunneling rate and is associated with the energy change by the value of *A* at low temperatures and proton jumps over the barrier *U_0_* at high temperatures. The solid curves show the dependence of the 1611 cm^−1^ mode of the C = O stretchings at low temperatures and are drawn according to the expression ω(*T*) = ω(0) − *C*<*n*>, where <*n*> is the average quantum number of phonon ω*_A_* and *C* is a proportionality constant (see [48]).

As can be estimated from Figure 32, the value of *A* is approximately equal to 12 meV in TPA(H) and to 9 meV in TPA(D). The constant *C* is 2.2 cm^−1^ in TPA(H) and 0.75 cm^−1^ in TPA(D). The latter fact suggests that the probability of proton tunneling is significantly higher than that of deuteron, which is in agreement with theoretical concepts. A slight linear increase of the ω_C = O_ at T > 100 K in both compounds (dashed straight lines in Figure 32) is the result of weakening of the hydrogen τ-bond at elevated temperatures and gradual termination of tunneling. It testifies that the hopping rate is low in TPA(H) and even lower in TPA(D). Possibly, no jumps take place in the latter case at all. It should be noted that the above numerical parameters of the barrier *A* and the fact that no jumps with high rates occur at elevated temperatures disagree with previous data [37], where *A* and the barrier *U_0_* were reported to be 16.25 MeV (130 сm^−1^) and 75 MeV (600 cm^−1^), respectively.

The Raman spectra of the compounds with tautomeric O–H···O hydrogen bonding in the temperature range 5–300 K perfectly illustrate the behavior of protons on the τ-bonds. The spectra allow one to separate in temperature different mechanisms of proton density distribution, and the study of hopping and tunneling processes becomes possible with the help of a simple and available Raman spectroscopy technique. The results presented in this work largely coincide with those reported previously [32,33,34,35,36,37,38,39,40,41,42]. However, we believe that the description of proton behavior in τ-bonds obtained here using Raman spectroscopy complements the knowledge obtained from the works based on nuclear magnetic resonance (NMA) and inelastic neutron scattering (INS) methods. Undoubtedly, the two latter methods provide highly reliable data, but they are capable to fix only the final phase of the tunneling process associated with proton transfer between the wells. In addition, at very low temperatures, only the more stable tautomer is noticeably populated, the protons are arranged in an ordered way, and the tunneling rate is low. In this case, the NMR *T*_1_ relaxation time is too large to be registered. The INS signal also becomes very weak. Therefore, none of these techniques can be used appropriately to study the dynamics at very low temperatures [36].

Also, the change in the energy of the process (in this case, the vibration frequency), which can be fixed in the Raman experiment, is of the order of tenths of an inverse centimeter (hundredths of millielectronvolts), which seems to be beyond the capabilities of NMA and INS.

## 6. Brief Characteristic of N–H···O and C–H···Y Hydrogen Bonds

### 6.1. N–H···O Hydrogen Bond

Despite the enormous importance of N–H···O hydrogen bonding in biological systems, their properties have not yet been fully established.

The dependence of the ν_N–H_ wavenumber on the N···O distance, known until recently [50], turned out to be significantly weaker than the analogous dependence for the O – H···O bond, and is shown in Figure 33 by straight line 1. This dependence is characterized by the slope χ = Δν/Δ*d*_N···O_ = 1400 cm^−1^/Å, i.e., even slightly less than for the weak O–H···O bond. The range of both *d*_N···O_ and ν_N–H_ variation is also relatively narrow.

The above dependence is fulfilled for most chemical compounds of various types with N–H···O bond. It is possible, however, to synthesize the compounds in which the N–H···O bond becomes very strong [51]. In a chain of conjugated bonds with an embedded N-H···O hydrogen bond, for example,
–C = C–N–H···O = C–C= ↔ =C–C = N···H–O–C = C–, 
ordinary C–C, and double C = C cease to exist as different bonds, and a intermediate one-and-a-half bond is established between the carbon atoms (this situation is also often realized for N–H···O embedded in the aromatic ring). This phenomenon in chemistry is called bond resonance. In the case, the hydrogen atom in N–H···O should also occupy some intermediate position between the N–H···O and N···H–O states, i.e., to be equidistant from both atoms, nitrogen and oxygen, which is a condition for the occurring of a strong bond. Figure 33 curve 2 shows the found ν_N–H_ wavenumber vs. N···O distance in the hydrogen bond N–H···O for conjugated systems [51]. This dependence is already similar to that for the O–H···O bond.

Dependence 2 (Figure 33) indeed shows a sharp decrease in the ν_N–H_ frequency at *d*_N···O_ < 2.6 Å, which is associated with an increase in hydrogen bonding. However, at 2.6 < *d*_N···O_ < 3.0 Å, the frequency of N–H stretchings in chains of conjugated bonds is not lower but even higher than the ν_N–H_ obtained in [50], which does not fit into the concept of hydrogen bond strengthening proposed by the authors of [51]. Finally, in the crystals of amino acids, N–H···O hydrogen bonds reveal another, different from 1 and 2, dependence of ν_N–H_ on the N···O distance (dashed line 3, Figure 33). In the latter case, for *d*_N···O_ in the range of 2.8 - 3.0 Å, the ν_N–H_ is significantly lower than in the previous two cases. Thus, the N–H···O hydrogen bond in compounds of various types demonstrates a significantly different functional dependence of the N–H vibrational frequency on the N···O distance. This makes it completely different from that of O–H···O bonds. 

There are two reasons for the fundamental difference between O–H···O and N–H···O hydrogen bonds. One of them, the main one, is that the nitrogen atom, in contrast to the oxygen atom, can have different oxidation states in chemical compounds. Indeed, the vibrational frequency of hydrogen-bonded N–H should substantially depend on the charge state of the nitrogen atom, and each of these states will have its own specific dependence of the ν_N–H_ on the N···O distance. Another reason, less obvious, relates directly to the mechanism of the hydrogen bond formation. Donor oxygen atoms in O–H···O use predominantly *p*-atomic orbitals to make O–H bonds, while donor nitrogen atoms in N–H···O use mainly hybridized *sp^n^*-atomic orbitals to bond with a hydrogen atom. In the latter case, the degree of participation of the *s*- or *p*-components will depend on the charge of the acceptor atom (i.e., in both cases, the acceptor oxygen) [52]. This is the Bent’s rule [53], according to which, in the X–H···Y hydrogen bond, the hybridized orbitals of the X atom have predominantly *s*-character in the case when they are directed to the electropositive substituent (i.e., Y), and predominantly *p*- character—in the case of an electronegative substituent. The character of the hybridized orbitals on the X–H···Y bond determines the ratio of the radii of the X and hydrogen atoms and, consequently, the length of the X–H bond. Thus, with the *s*-character of hybridized X orbitals, the X–H bond length decreases, and the ν_X–H_ vibrational frequency increases. For *p*-character, the opposite is true.

Thus, the ν_N–H_ frequency is not a good descriptor for the strength of the N–H···O hydrogen bond. This is the main difference between the N–H···O and O–H···O bonds. As mentioned above, the donor oxygen atom in O–H···O mainly uses only *p*-atomic orbitals for bonding with the hydrogen atom in any type of compound. For this reason, there may be just one dependence of ν_O–H_ on *d*_O···O_ in O–H···O hydrogen bonds, which manifests itself in a wide range of bond lengths and vibrational frequencies (Figure 9).

### 6.2. Weak C–H···Y Bonds. “Blue” Shift

Hydrogen C–H···Y bonds are very weak, their energy of interaction is comparable to the energy of van der Waals bond, but the spectroscopic manifestation of these bonds has been reliably established. In the literature, however, there is no information on the experimental dependence of the ν_C–H_ stretching vibrations on the C···Y distance; similar to those that exist for O–H···O and N–H···O bonds. The reason for this is not so much the smallness of the effect, but rather the unusual properties of the C-H···Y hydrogen bond. The fact is that along with the usual shift of the ν_C–H_ to the low-frequency region during the occurrence of the C–H···Y hydrogen bond (“red” shift), sometimes ν_C–H_ is shifted to the high-frequency region (“blue” shift). This phenomenon remained unclear for a long time, and only in the last decade did a quantum-chemical model appear, which satisfactorily explains the nature of the “blue” shift [52].

The reason for the unusual behavior of the C–H···Y bond was the carbon atom, which, unlike other donor atoms, actually uses the hybridized atomic *sp*^3^ orbitals in the formation of valence interactions with the neighbors. In this case, the Bent rule, which was discussed in the previous paragraph, comes into force. Further, taking into account the weakness of the C–H···Y bond in general, the effect of the Bent rule can shift the ν_C–H_ frequency to both high-frequency and low-frequency regions. 

## 7. Conclusions

As shown above, an extremely strong hydrogen bond can occur under two conditions: (1) Complete identity of the donor and acceptor atoms and (2) high electronegativity of both the donor and acceptor. From this point of view, the hydrogen O–H···O bond seems to be almost ideal for the realization of a strong bond. Another common bond in chemistry, N–H···N, yields to O–H···O in the second condition—the electronegativity of nitrogen atoms. On the other hand, the strongest hydrogen bond that exists in nature is the symmetric F···H···F, the energy of which is explained precisely by the very high electronegativity of fluorine atoms. However, the F···H···F bond is more of a curiosity in chemistry in terms of its prevalence than the subject of serious research.

A strong O–H···O hydrogen bond can be realized as an intramolecular bond built into the aromatic ring. An excellent overview of this problem was given in the recent work [54]. However, the study of Raman scattering of such compounds is a separate topic and is planned for the future.

The electronegativity of oxygen-acceptor is the parameter that sets the rigidity of a hydrogen bond in a wide range, from weak to extremely strong. It is the low or zero electronegativity of oxygen atoms that can explain the fact that in many framework silicates or aluminosilicates, a water molecule included in the crystal cavity practically does not interact with the neighbor oxygen atoms. In framework crystals, oxygen atoms forming a cavity in the lattice are, as a rule, bridging between two silicon atoms. The valence orbitals of such oxygen atoms are saturated, and they do not show a tendency to interact with the hydrogen atoms of the H_2_O molecule included in the cavity.

Finally, the O–H···O hydrogen bond very often acts as the base of another extremely interesting and important phenomenon in chemistry—tautomerism. Tautomeric O–H···O hydrogen bonds allow us to investigate complex processes of proton dynamics on a hydrogen bond—tunneling and jumping, significantly expanding our knowledge of the behavior of quantum particles such as the proton.

## 8. Experimental

Glycinium phosphite and protonated di-methylformamid were chosen as compounds with strong O-H···O hydrogen bonding. The crystals of glycinium phosphite, NH_3_CH_2_COOH H_2_PO_3_, were prepared by N. Bogdanov (Novosibirsk State University, Russia) and protonated di-methylformamid [(DMF)_2_H]_2_[W_6_Cl_14_] was prepared by P. Abramov (Nikolaev Institute of Inorganic Chemistry, Novosibirsk, Russia). The synthesis and structure of this compound were reported in Ref. [30].

The Raman spectra were collected on a LabRAM Horiba single stage spectrometer with a CCD Symphony (Jobin Yvon) detector with 2048 horizontal pixels. The laser power (633 nm line of He-Ne laser) at the sample was typically less than 0.1 mW. The spectra at all temperatures were measured in backscattering collection geometry with a Raman microscope. The powder was fixed on the cold finger of the cryostat. The spectral resolution was 0.7 cm^−1^.

## Figures and Tables

**Figure 1 ijms-22-05380-f001:**
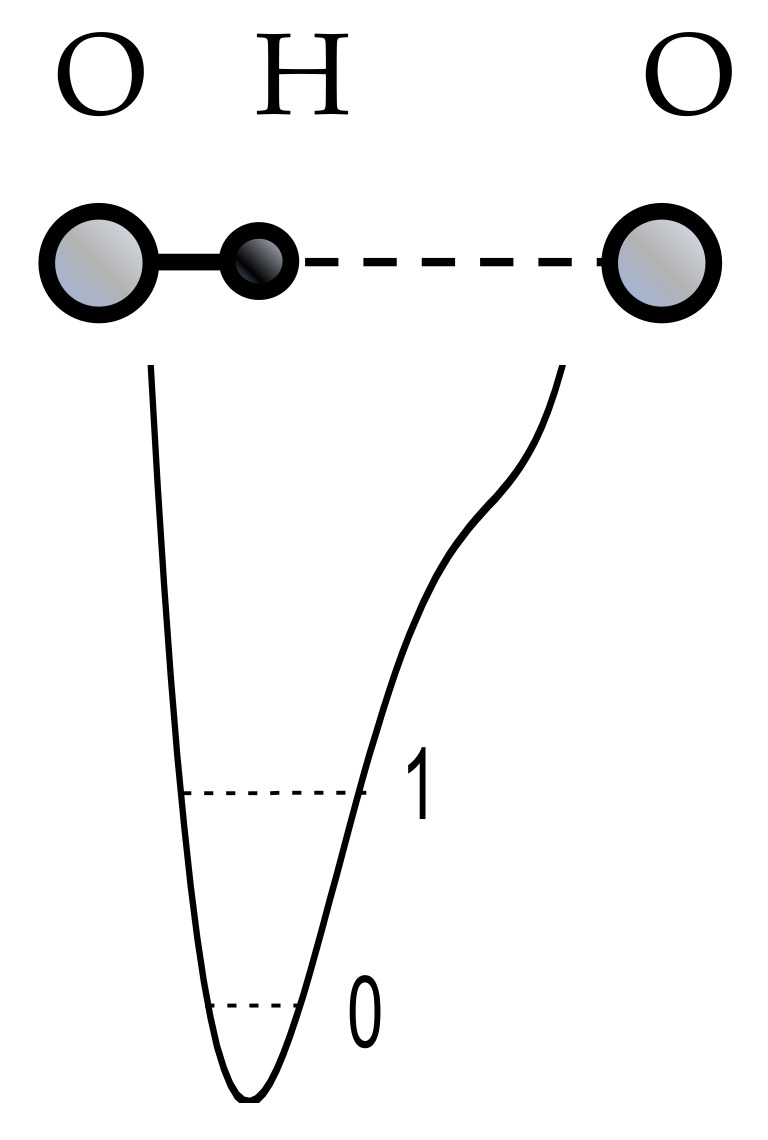
Proton potential function for a weak hydrogen bond, *d*_O···O_ ≥ 2.7 Å.

**Figure 2 ijms-22-05380-f002:**
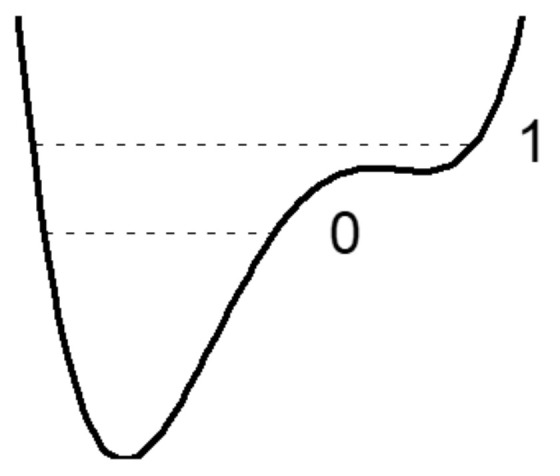
Proton potential function for a moderate hydrogen bond, *d*_O···O_ ~ 2.6 Å.

**Figure 3 ijms-22-05380-f003:**
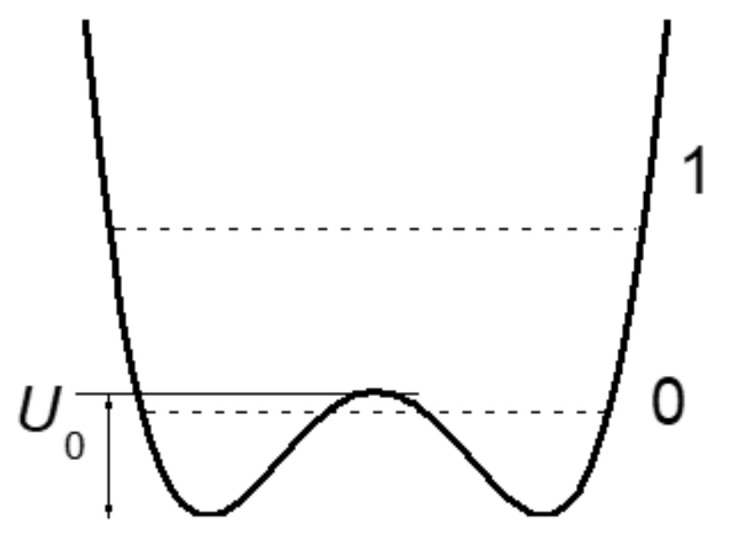
Proton potential function for a strong hydrogen bond (“deep tunneling regime”), *d*_O···O_ ~ 2.5 Å.

**Figure 4 ijms-22-05380-f004:**
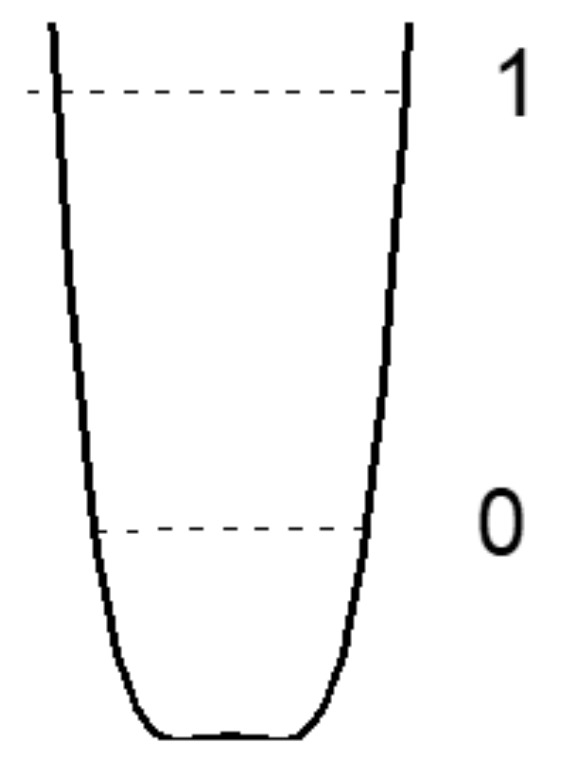
Proton potential function for an extremely strong hydrogen bond (“ultrashort, centered HB”), *d*_O···O_ < 2.3 Å.

**Figure 5 ijms-22-05380-f005:**
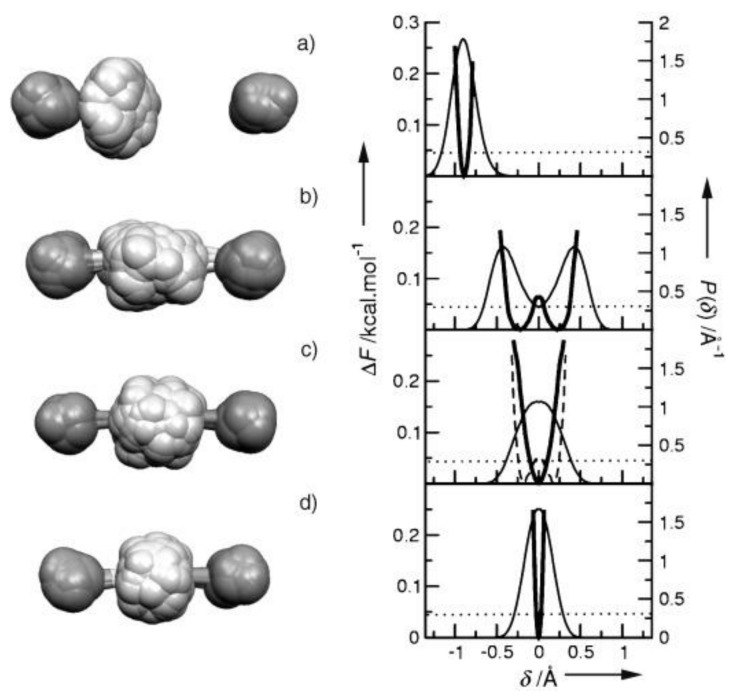
Calculation [17] of the distribution function of proton density (left) and potential energy of a proton (right) depending on the length of the O–H···O bond. (**a**) weak bond, *d*_O···O_ = 2.85 Å, (**b**) strong bond, *d*_O···O_ = 2.43 Å, (**c**) strong bond, *d*_O···O_ = 2.29 Å, (**d**) extremely strong bond, *d*_O···O_ = 2.17 Å. δ = *R*_OaH_ − *R*_ObH_. (See also Figure 1 in [17]). The horizontal dotted line indicates the thermal energy level *kT* at 25 K. The figure is reproduced with permission from Ref. [17], copyright (2005) European Chemical Societies Publishing.

**Figure 6 ijms-22-05380-f006:**
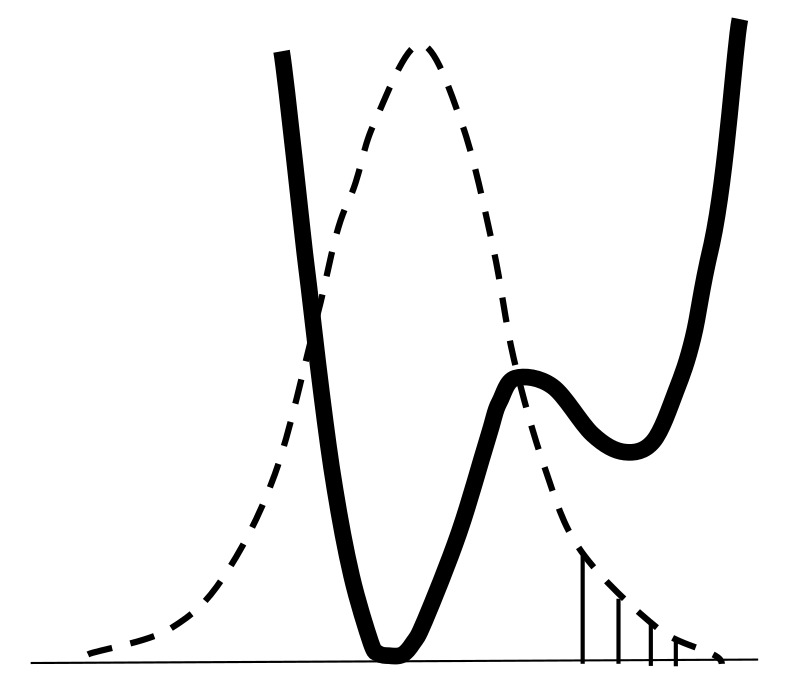
The assumed proton density distribution (dotted curve) at *d*_O···O_ < 2.6 Å. The region of the proton density spread into the adjacent well is shaded.

**Figure 7 ijms-22-05380-f007:**
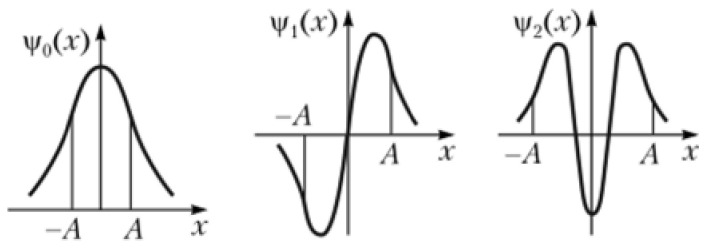
Wave functions of the harmonic oscillator. Vertical dashed lines indicate the turning points of the classical oscillator.

**Figure 8 ijms-22-05380-f008:**
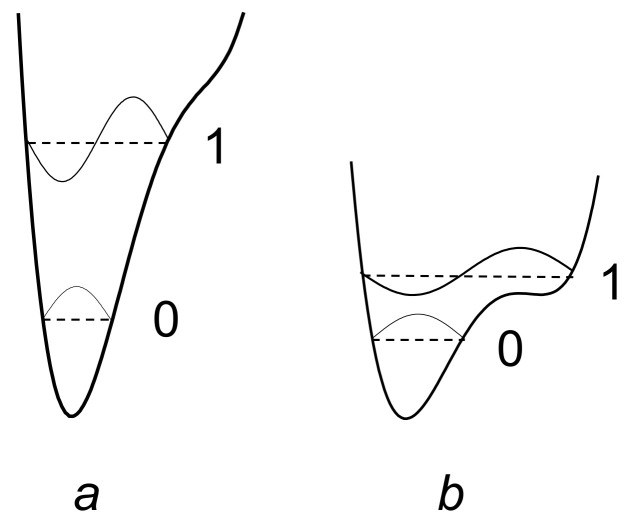
Schematic representation of the ground and excited vibrational states for a weak (**a**) and moderate (**b**) bonds.

**Figure 9 ijms-22-05380-f009:**
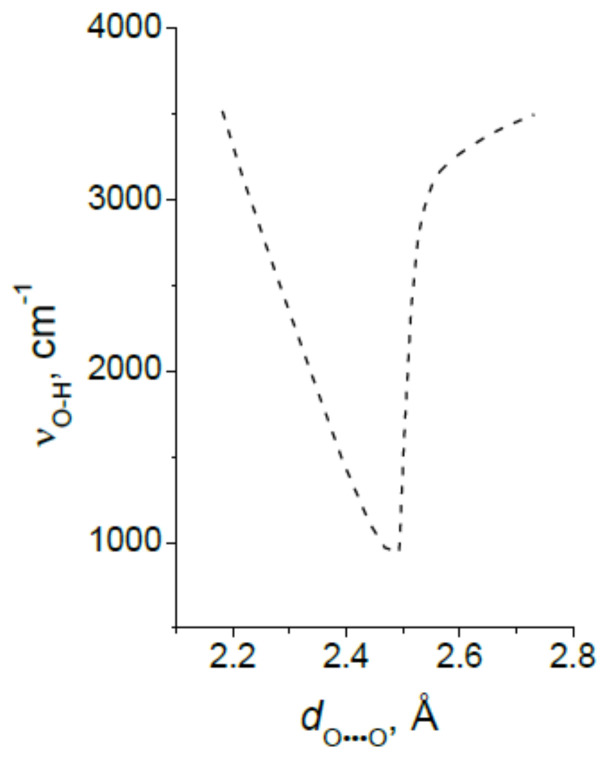
Assumed dependence of O–H vibrational frequency as a function of distance *d*_O···O_ for O−H···O hydrogen bonding.

**Figure 10 ijms-22-05380-f010:**
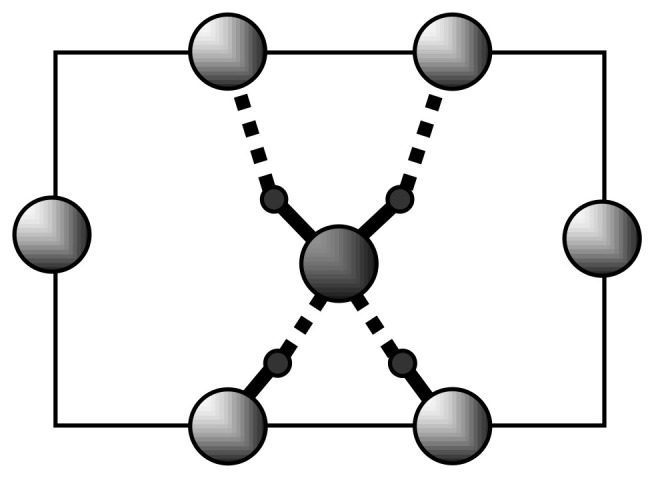
The system of hydrogen bonds of the H_2_O molecule (in the center) and the OH groups in the cavity of the hemimorphite mineral.

**Figure 11 ijms-22-05380-f011:**
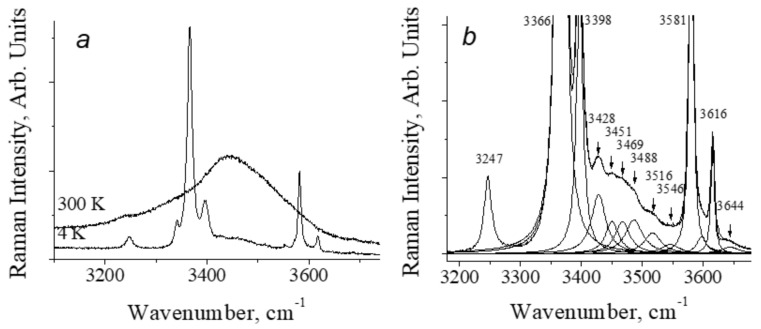
(**a**) Spectra of the stretching vibrations of H_2_O molecules and OH groups in the hemimorphite cavity at 4 K and 300 K. The H_2_O molecule in the cavity forms a hydrogen bond with the OH groups of the crystal as a donor (lines ~3400 cm^−1^ at 4 K) and as an acceptor (~3600 cm^−1^); (**b**) Spectrum (4 K) of the combined modes of the stretching vibrations of H_2_O molecules in the cavity of hemimorphite with translational modes of H_2_O itself and of the lattice of the host crystal [22].

**Figure 12 ijms-22-05380-f012:**
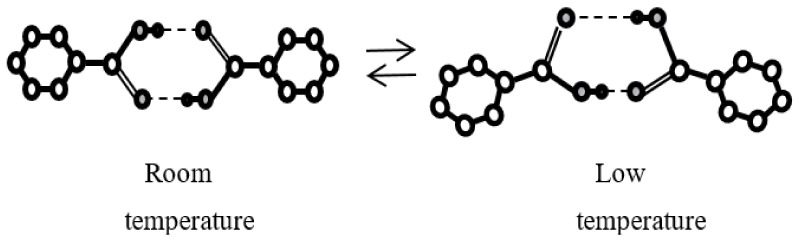
Dimers of benzoic acid in a crystal at room and low temperatures.

**Figure 13 ijms-22-05380-f013:**
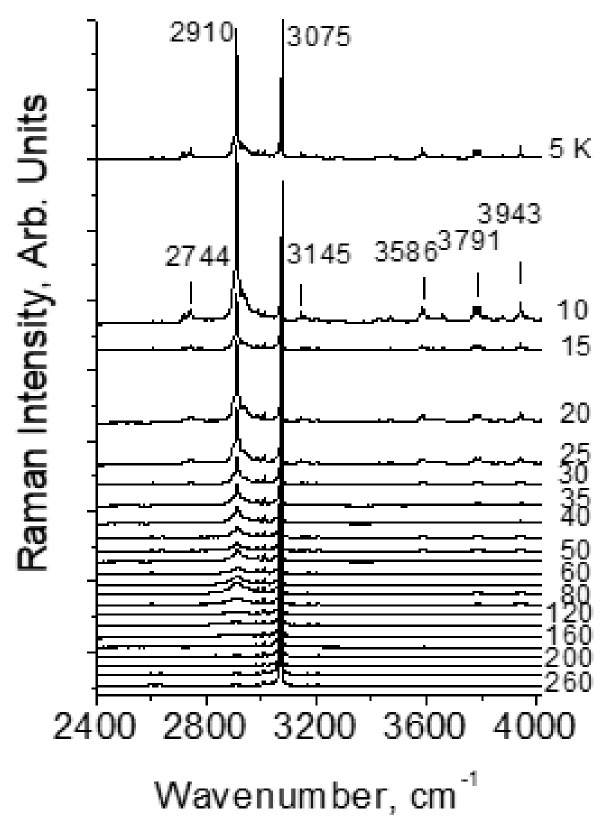
Raman spectra of crystals of benzoic acid in the high-frequency region at different temperatures [24].

**Figure 14 ijms-22-05380-f014:**
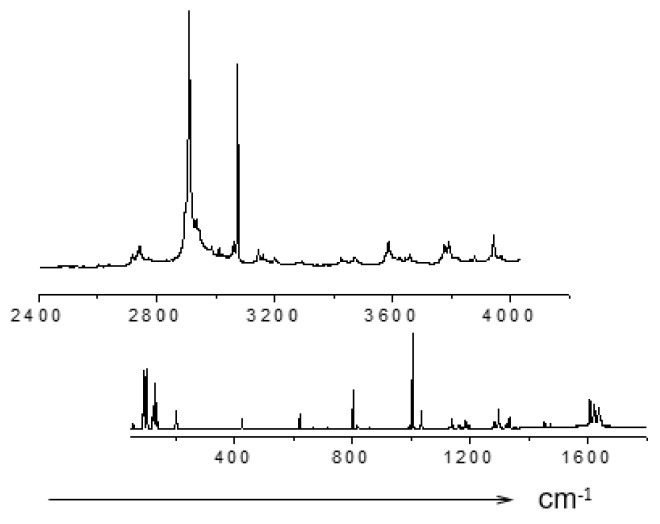
A comparison of the high-wavenumber (2400–4000 cm^−1^, top) and intramolecular (50–1700 cm^−1^, bottom) vibrations at T = 10 K.

**Figure 15 ijms-22-05380-f015:**
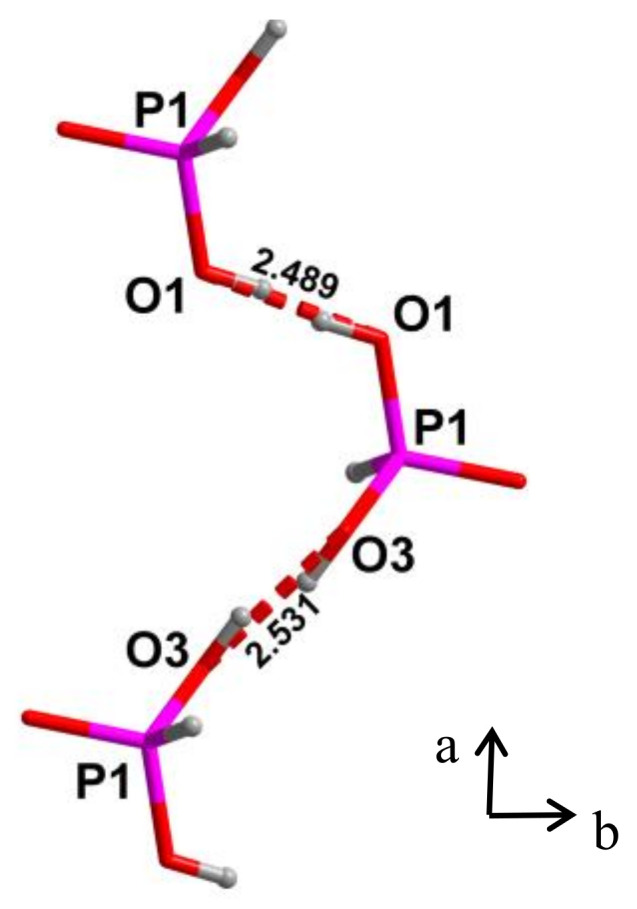
Fragment of the crystal structure of glycine phosphate.

**Figure 16 ijms-22-05380-f016:**
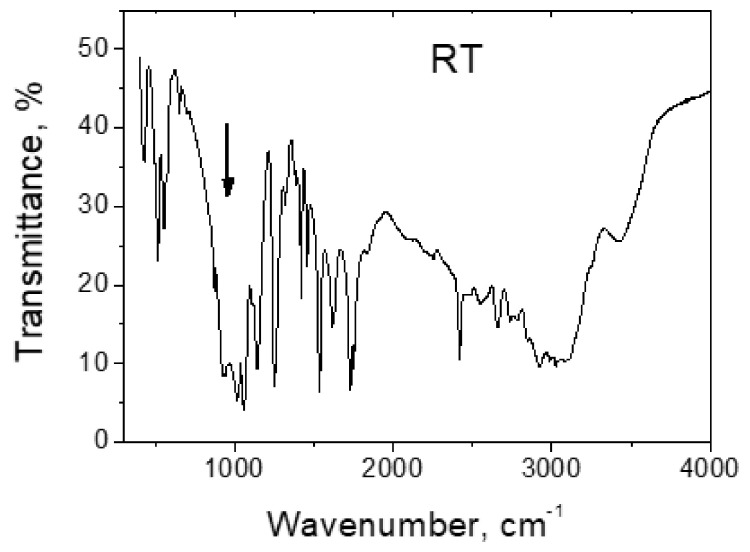
IR-spectrum of glycinium phosphite at room temperature. The arrow shows the expected position of the hydrogen bond absorption band. The Raman band in this region appears only at low temperatures (see Figure 17a).

**Figure 17 ijms-22-05380-f017:**
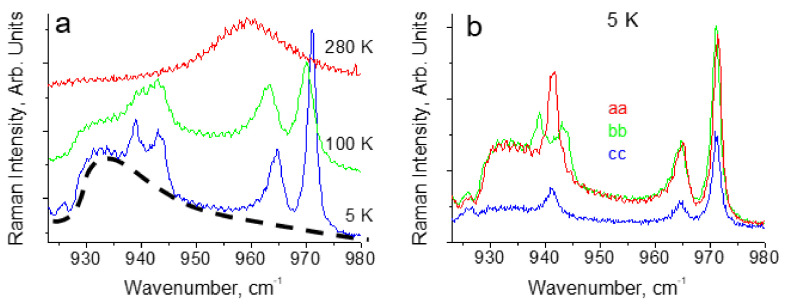
(**a**,**b**) Raman spectra of glycinium phosphite in the range of the hydrogen bond vibrations at different temperatures (**a**) and orientation of the polarization vector of the incident and scattered light relative to the crystallographic directions (**b**).

**Figure 18 ijms-22-05380-f018:**
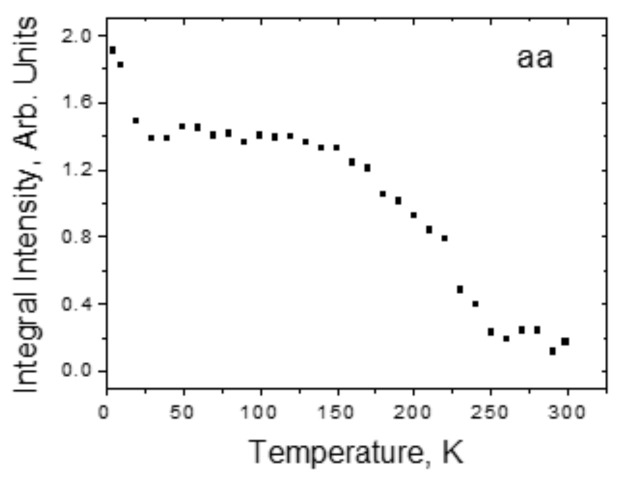
Relative integral intensity of the 930–980 cm^−1^ band as a function of crystal temperature.

**Figure 19 ijms-22-05380-f019:**
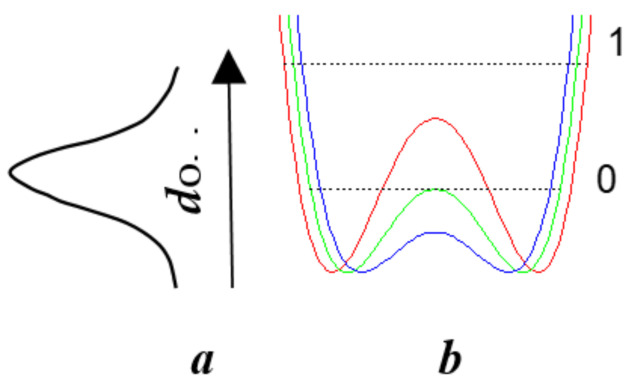
(**a**,**b**) The occurrence of an unusual contour of the proton vibrational band. (**a**)—distribution of *d*_O__···O_ distances, arising from the quantum uncertainty of the coordinate of oxygen atoms; (**b**)—energy of zeroth motions relative to the barrier *U*_0_ for short (blue curve), medium (green curve) and long (red curve) *d*_O__···O_ distances.

**Figure 20 ijms-22-05380-f020:**
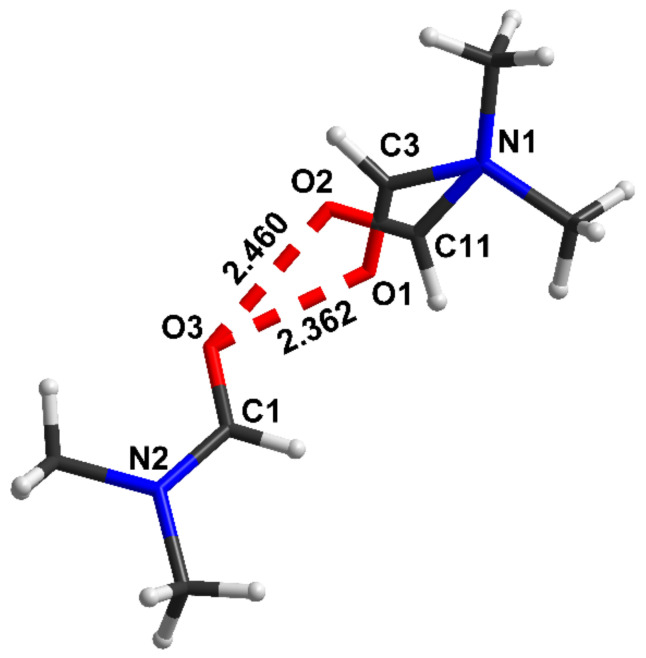
The structure of [(DMF)_2_H.

**Figure 21 ijms-22-05380-f021:**
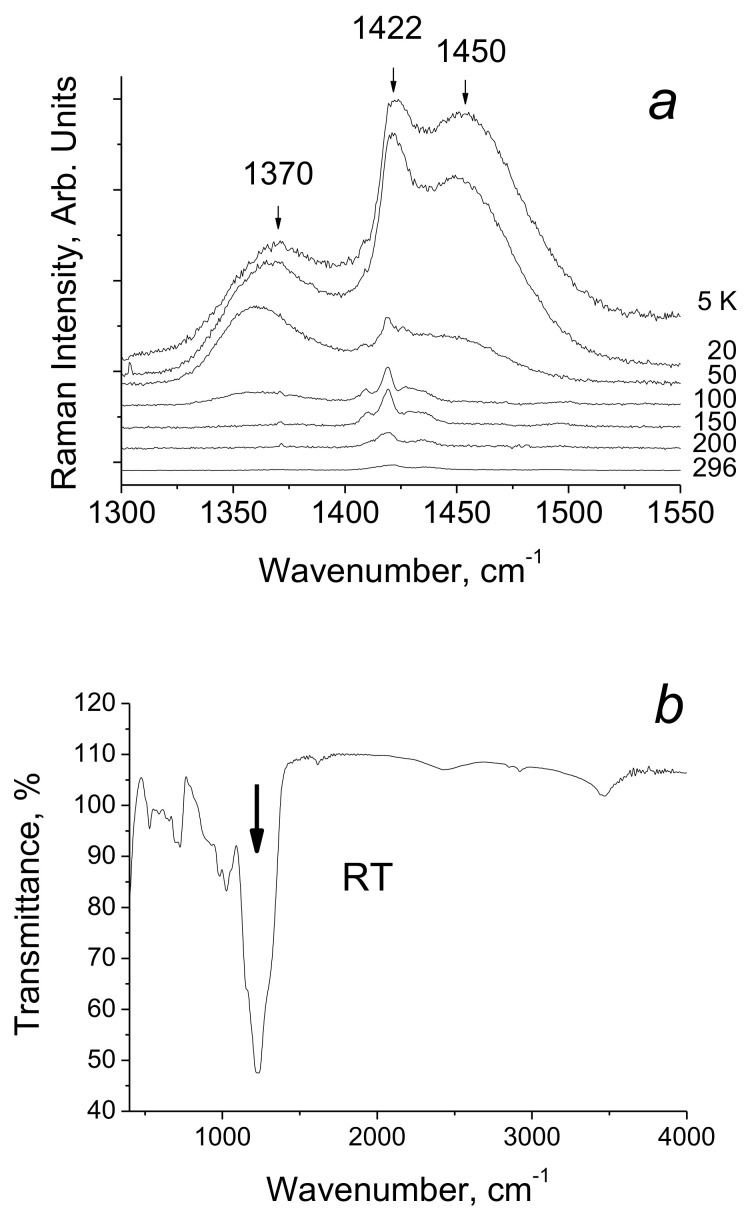
(**a**) Raman spectra of [(DMF)_2_H]_2_[W_6_Cl_14_] in the region of proton vibrations at various temperatures. (**b**) IR-spectrum of [(DMF)_2_H]_2_[W_6_Cl_14_] at room temperature [30]. The arrow shows the expected position of the hydrogen bond absorption band. The corresponding Raman bands in this region rise only at low temperatures (**a**).

**Figure 22 ijms-22-05380-f022:**
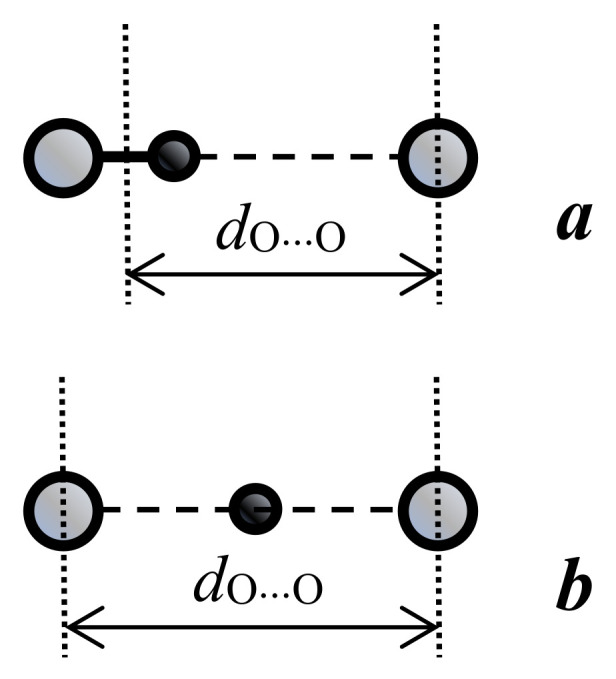
X-ray measurement of *d*_O__···O_ distance at localization of a proton near the donor (**a**) and in the midpoint of the donor-acceptor bond (**b**). The true distance between the centers of oxygen atoms in the figure is the same.

**Figure 23 ijms-22-05380-f023:**
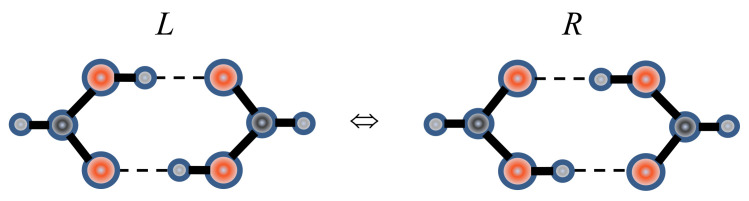
*L*- and *R* tautomers in dimer of formic acid.

**Figure 24 ijms-22-05380-f024:**
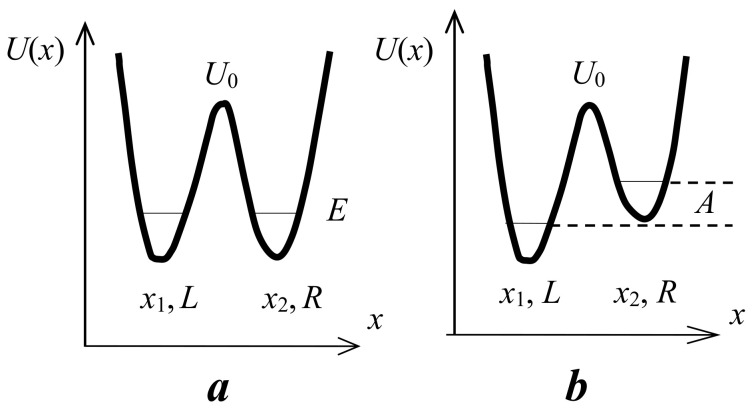
(**a**,**b**) Potential energy of the proton on the τ-bond of an isolated dimer (trimer, etc.) in the gas phase (**a**), the same energy of a real dimer in the crystal (**b**). *E* is the particle energy in the potential well, *U*_0_ is the barrier height, (*x*_1_, *x*_2_) are the coordinates of potential energy minima. *A* is the difference between the energies of *L*- and *R*-tautomers in the crystal lattice.

**Figure 25 ijms-22-05380-f025:**
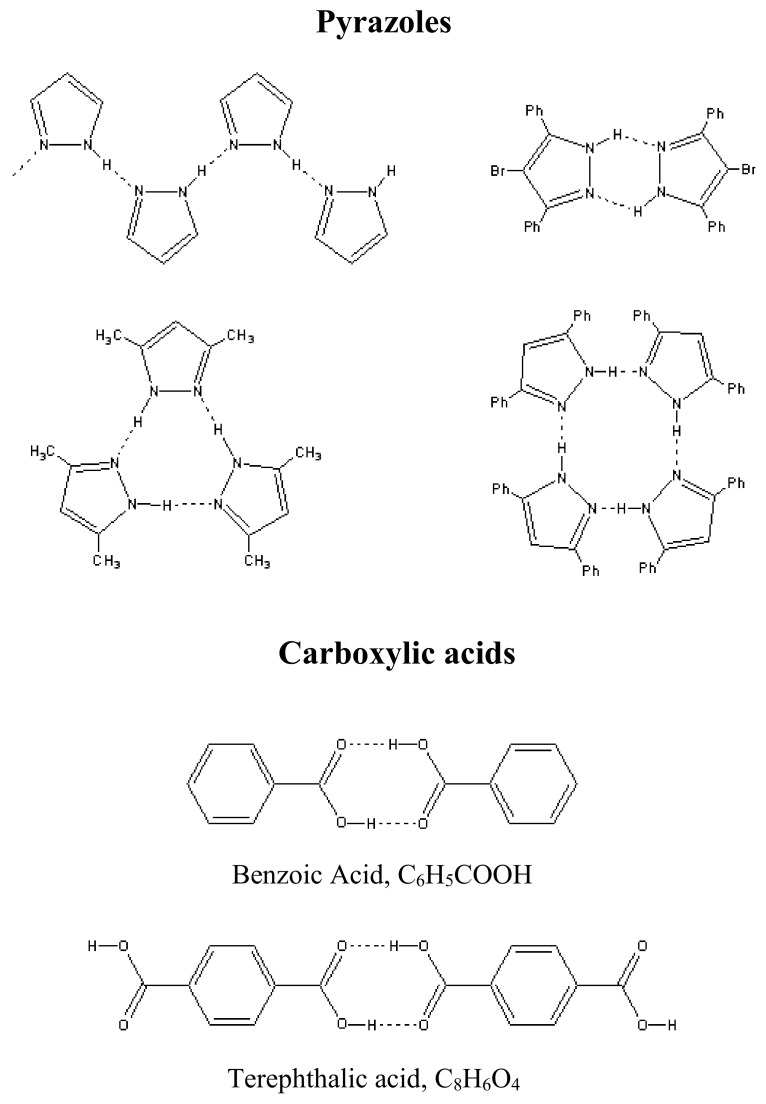
An example of compounds with τ-bond.

**Figure 26 ijms-22-05380-f026:**
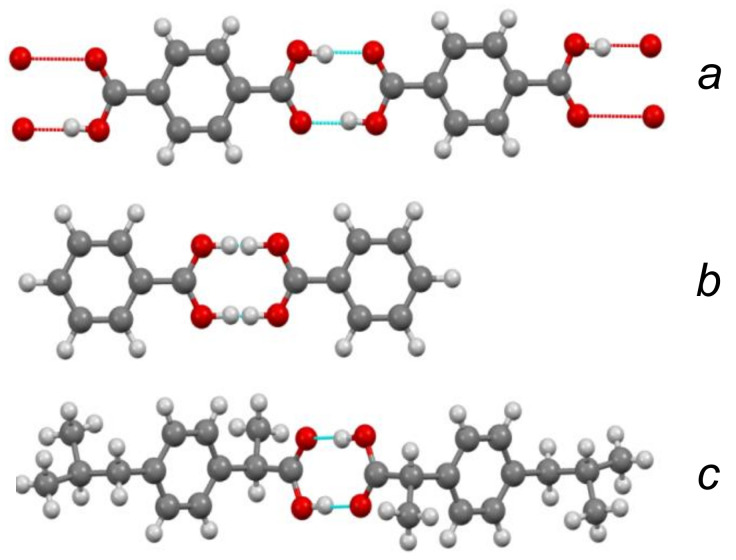
Chains of TPA molecules with symmetric- (**a**), BZ dimers with quasi-symmetric- (**b**), and IB dimers with asymmetric τ-bond (**c**). A slight deviation from symmetry in the benzoic acid dimer arises due to the crystalline effect, in which one of the four oxygen atoms of the τ-bond has a shorter contact with the environment than the other three. The asymmetry of dimer IB is due to two different carbon substituents adjacent to the α-carbon of the hydroxyl group.

**Figure 27 ijms-22-05380-f027:**
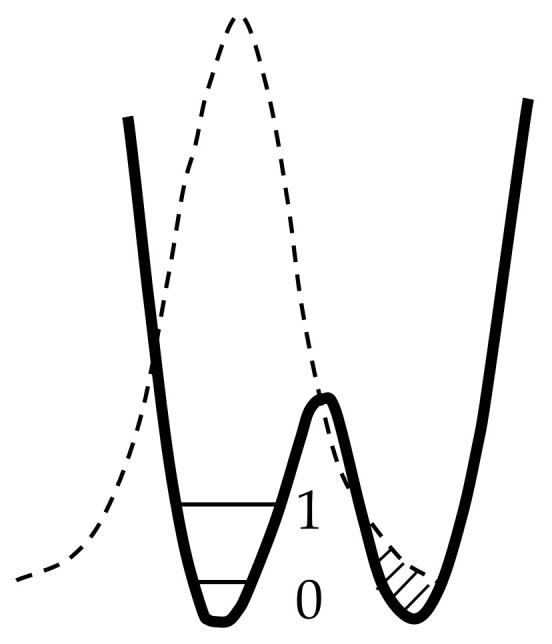
The assumed distribution of the proton density (dashed curve) at *d*_O···O_ < 2.6 Å. The shaded area shows the region where the proton density penetrates into the neighboring well.

**Figure 28 ijms-22-05380-f028:**
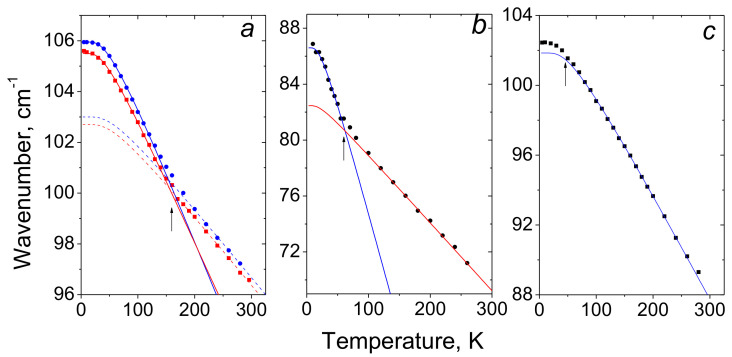
ω_τ_ as function of temperature in symmetric TPA(H) (filled circles) and TPA(D) (filled squares) (**a**), quasisymmetric BZ (**b**), and asymmetric IB (**c**) tautomeric bonds. Solid curves are plotted from the anharmonic shifts of the phonon ω determined by its thermal population [48]. The arrow marks the position of the breakpoint of ω_τ_(*T*).

**Figure 29 ijms-22-05380-f029:**
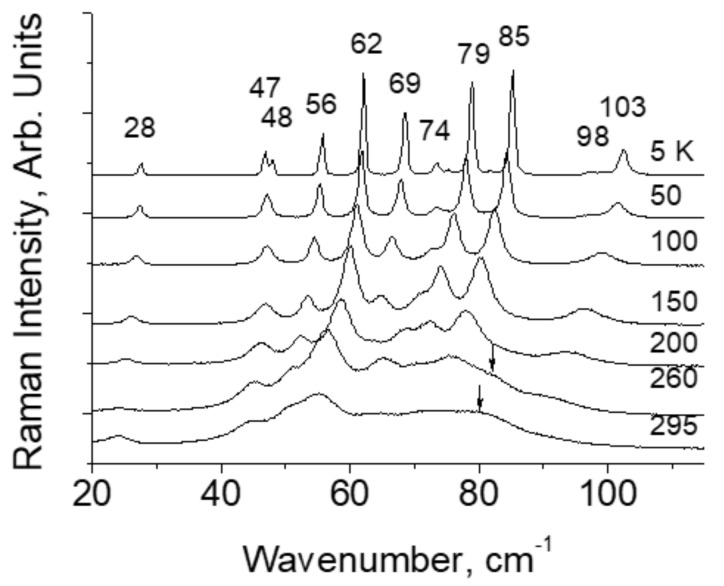
Low-frequency spectra of ibuprofen crystals at different temperatures [43]. The arrow marks the position of a new high-temperature mode at ~90 cm^−1^ arising at Т ≥ 150 K.

**Figure 30 ijms-22-05380-f030:**
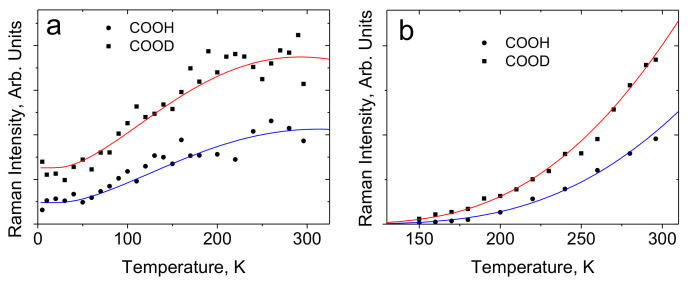
(**a**,**b**) Temperature dependence of the integral intensity of ~100 cm^−1^ mode (**a**) and ~90 cm^−1^ mode (**b**) in normal IB (H) and deuterated IB (D) crystals. The ΔE value is 80 meV for IB (H) and 70 meV for IB (D). More details in [43].

**Figure 31 ijms-22-05380-f031:**
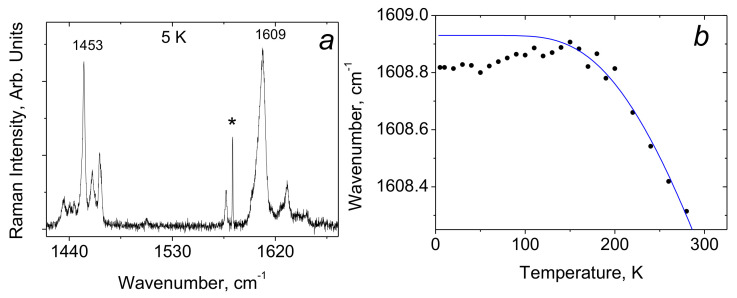
(**a**) The spectrum of ibuprofen (COOH) at 5 K in the range of C = O vibrations. The asterisk marks the position of line of Ne lamp, which was registered at each temperature for correction of the spectra. (**b**) Temperature dependence of the position of the 1609 cm^−1^ band peak of the C = O stretching vibration. The solid curve is plotted under the condition that a change in the mode frequency is caused by an activation process with Δ*E* = 80 meV.

**Figure 32 ijms-22-05380-f032:**
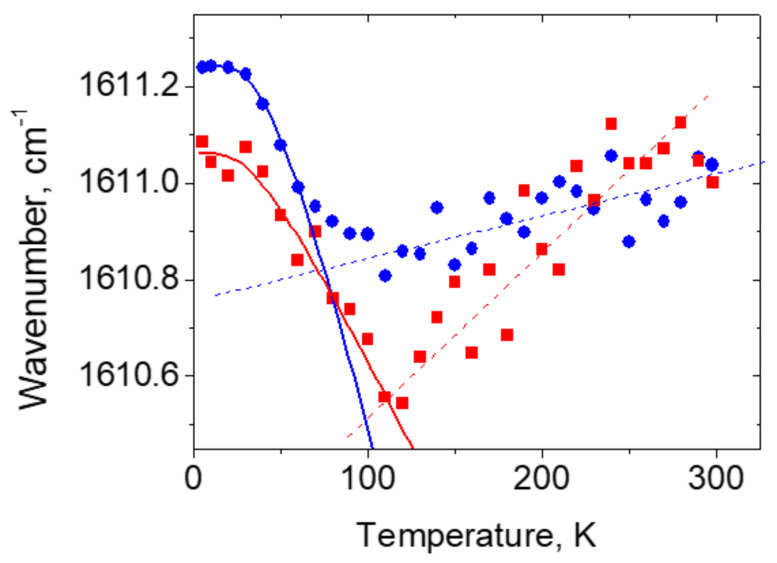
Temperature dependence of the C = O vibrational wavenumber of TPA(H) (circles) and TPA(D) (squares) carboxylic groups. Solid curves show temperature dependences of the thermal population of phonons ω*_A_* (see the text); the dependences above 100 K are imaged by dashed lines, which are drawn approximately.

**Figure 33 ijms-22-05380-f033:**
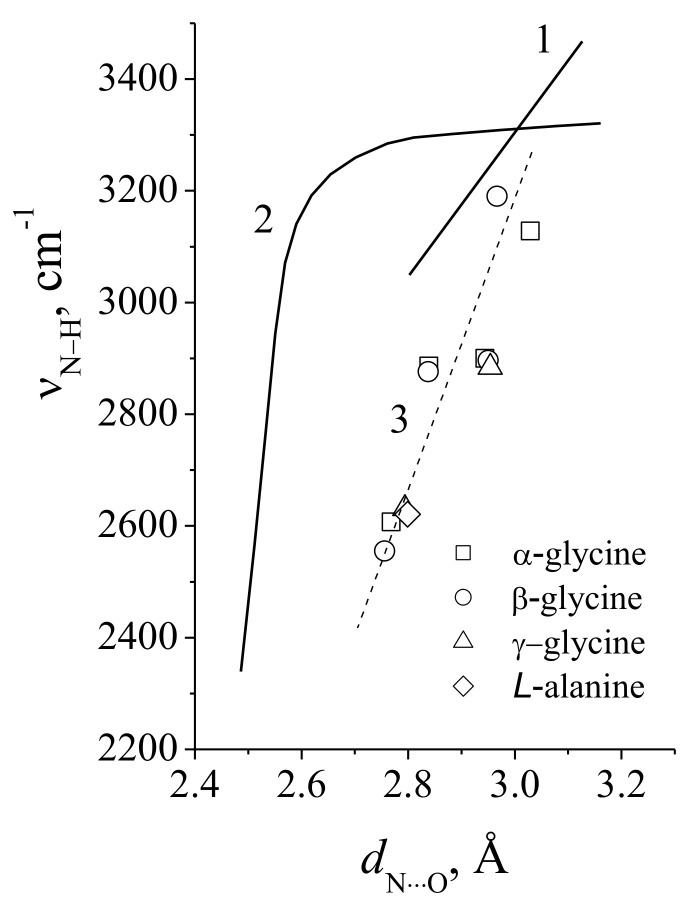
ν_N–H_ stretching vibrations in the N–H···O hydrogen bond as function of N···O distance. 1 – N–H···O in compounds of various types [50], 2 – N–H···O bonds in conjugated systems [51], 3 – N–H···O in amino acid crystals.

**Table 1 ijms-22-05380-t001:** The length of the hydrogen bond and the frequency of translational vibrations ω_τ_ in the studied compounds.

Composition	*d*_O···O_, Å	ω_τ_, cm^−1^	
		Calculation [43,44]	Experiment, 5 K
TPA(H)	2.62 [45]	114	106
TPA(D)		113	105
BZ	2.63 [46]	–	86
IB(H)	2.66 [47]	108	104
IB(D)		–	103

## Data Availability

No data availability statement.

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
