# Peer review of "Hydrogen Bonds: Raman Spectroscopic Study"

_ijms, 2021, doi:10.3390/ijms22105380_

Round 1

Reviewer 1 Report

Recently, a lot of similar review articles on hydrogen bond have been published, i.e. those authored by Scheiner or Grabowski. The topic of this review is more suitable for a book chapter rather than review that in its nature should be concise and focused on a particular topic. However, this particular review is interesting, well written and can be published after revisions listed below.

The title is not very informative. I am also a fan of short titles but please make it a little bit longer.

Lines 18-24, I don’t think that such a classification is justified. I.e. what is the difference between “ionic” and “Coulomb” interactions? Further, what about London forces? Besides, aren’t van der Waals “chemical”?

Line 29, again, what about London forces?

Line 30, this strong statement needs a support, i.e. in the number of articles. Personally, I can’t agree with this.

Line 33, I can’t agree again. Calculations can provide much more information, i.e. NMR properties, thermodynamic properties, dynamics, optical properties etc.

Line 43, it should be “broad”

Line 47, what about NMR?

Line 127, Is it somehow related with the increase of entropy?

In the article the type of hydrogen bond involving tritium is not mentioned at all and I guess it should be.

In the whole manuscript the NMR studies of H bond are only mentioned twice. This is a powerful analytical technique that can give the information unobtainable by any other spectroscopic method. I think the Authors should provide more information on the NMR studies, not only 1H but also on other nuclei.

Why most of the figures are shifted to the left? They should be in the middle of the page.

Line 212, can you please proved some examples of such system?

Figure 9 and the discussion lines 385-420, I think this work may be helpful : DOI: 10.3390/molecules22040552  (I am NOT an author of this work but I think it may be useful to include it in this review).

Line 596, “Electronegativity  of  oxygen-donor  and  oxygen-acceptor.” – this should be the title of the paragraph.

Line 637, what do you mean by “crystalline effects”?

There are two figures numbered “12”.

Line 865, there is a typo here in the word “than”

Figure 20 is cut in the middle.

There has been an enormous increase in the quality of the results received from quantum mechanical calculations. I think the Author should at least briefly discuss the finding on hydrogen bonds received from ab initio molecular dynamics simulations.

Lines 1014-1064, an what if one of those hydrogen atom would be deuterium? This would make the situation even more complicated but a the same time more interesting. Have you found any theoretical (or experimental) studies on such systems?

Line 1185, why plural?

Line 1498, a reference is needed here.

Finally, the number of references is very limited, taking into consideration the length of this review.

Author Response

Referee 1
1. The title is not very informative. I am also a fan of short titles but please make it a little bit longer.

Done

  1. Lines 18-24, I don’t think that such a classification is justified. I.e. what is the difference between “ionic” and “Coulomb” interactions? Further, what about London forces? Besides, aren’t van der Waals “chemical”?

“ionic” and “Coulomb” interactions are different certainly. London forces is a special case of van der Waals interaction. Van der Waals interaction is an interaction of dipole momentums (i.e. physical), not an interaction of atomic orbitals (i.e. chemical).

  1. Line 29, again, what about London forces

See previous comment.

  1. Line 30, this strong statement needs a support, i.e. in the number of articles. Personally, I can’t agree with this.

Yes I agree. A change is made in the text.

Line 33, I can’t agree again. Calculations can provide much more information, i.e. NMR properties, thermodynamic properties, dynamics, optical properties etc.

Yes, of course, if you set such a task to the calculation. But I meant the calculation of the vibrational spectrum of compounds.

  1. Line 43, it should be “broad”

Done.

  1. Line 47, what about NMR?

I am not an expert in NMR, therefore, these issues are not addressed in the work.

  1. Line 127, Is it somehow related with the increase of entropy?

I think yes, but neither Marks ]17] nor Wang [18] discussed this issue in their works.

  1. In the article the type of hydrogen bond involving tritium is not mentioned at all and I guess it should be.

Yes, this is true, but, unfortunately, compounds with tritium were not available to the author. Therefore, the work does not discuss the spectra of tritium-substituted hydrogen bonds.

  1. In the whole manuscript the NMR studies of H bond are only mentioned twice. This is a powerful analytical technique that can give the information unobtainable by any other spectroscopic method. I think the Authors should provide more information on the NMR studies, not only 1H but also on other nuclei.

Discussion of NMR spectra is beyond the scope of this work.

  1. Why most of the figures are shifted to the left? They should be in the middle of the page.

I hope that the technical editor of the journal will solve this problem.

  1. Line 212, can you please proved some examples of such system?

There are probably examples of compounds with symmetric bonds in the literature, but I am deeply convinced that such statements need verification, including Raman spectra at low temperatures. Therefore, I cannot give examples, but I hope that the hydrogen bond in DMF2H, described in this work, is very close to symmetric.

  1. Figure 9 and the discussion lines 385-420, I think this work may be helpful : DOI: 10.3390/molecules22040552  (I am NOT an author of this work but I think it may be useful to include it in this review).

The work recommended by the reviewer is high quality and very interesting. But it refers to intramolecular hydrogen bonds, i.e. a topic that I deliberately avoided in my article, hoping that it will become the main one in the next stage of work. However, a link to the recommended article is inserted in the text (see "Conclusion").

  1. Line 596, “Electronegativity  of  oxygen-donor  and  oxygen-acceptor.” – this should be the title of the paragraph.

Done.

  1. Line 637, what do you mean by “crystalline effects”?

One of the four oxygen atoms of the t-cycle has an additional short contact with the environment in the crystal lattice. This remark has been inserted into the text.

  1. There are two figures numbered “12”.

Numbering of figures is corrected.

  1. Line 865, there is a typo here in the word “than”

Corrected.

  1. Figure 20 is cut in the middle.

Corrected.

  1. There has been an enormous increase in the quality of the results received from quantum mechanical calculations. I think the Author should at least briefly discuss the finding on hydrogen bonds received from ab initio molecular dynamics simulations.

I completely agree with the opinion of the reviewer and also believe that a deep discussion of the results of various experiments, including NMR, and theoretical calculations, including ab initio molecular dynamics simulations, would only beautify the work. However, it seems to me that these additions will turn the article into a monograph. In addition, entire concept of hydrogen bonding presented in my paper is based on a high level quantum chemical calculation made by the authors [17,18]. I hope the [17,18] results are not inferior to those mentioned by the reviewer.

  1. Lines 1014-1064, an what if one of those hydrogen atom would be deuterium? This would make the situation even more complicated but a the same time more interesting. Have you found any theoretical (or experimental) studies on such systems?

Of course, the effects of deuteration are also discussed in the work using terephthalic acid and ibuprofen as examples (see paragraph 5).

  1. Line 1185, why plural?

Paragraph numbering was omitted. A change is made in the text.

  1. Line 1498, a reference is needed here.

Done.

23.Finally, the number of references is very limited, taking into consideration the length of this review.

Yes, I agree, however, all the most necessary links are provided.

Reviewer 2 Report

The manuscript is well-organized and well-presented. I have only one recommendation.

Please try to include one section on the significance and application of hydrogen bonding  (for example solution structure, self-assembly, and nanomaterial synthesis).

Author Response

Referee 2

  1. Please try to include one section on the significance and application of hydrogen bonding  (for example solution structure, self-assembly, and nanomaterial synthesis).

Undoubtedly, the significance and importance of hydrogen bonds in chemistry, nature and living organisms is enormous, and the section devoted to this topic would only beautify this work. But it is unlikely that I will be able to write a deep and high-quality section on this topic in a very short time (one week), which the editorial staff of the journal provides to correct the article. However, a small insert about the significance of the hydrogen bond is made in the text (see "Introduction").

Round 2

Reviewer 1 Report

The Author has corrected the manuscript, following my suggestions. The current version can be accepted.